# Uncovering the structure of self-regulation through data-driven ontology discovery

Ian W. Eisenberg[1], Patrick G. Bissett[1], A. Zeynep Enkavi [1], Jamie Li[1], David P. MacKinnon[2], Lisa A. Marsch[3] & Russell A. Poldrack[1]

Psychological sciences have identified a wealth of cognitive processes and behavioral phenomena, yet struggle to produce cumulative knowledge. Progress is hamstrung by siloed scientific traditions and a focus on explanation over prediction, two issues that are particularly damaging for the study of multifaceted constructs like self-regulation. Here, we derive a psychological ontology from a study of individual differences across a broad range of behavioral tasks, self-report surveys, and self-reported real-world outcomes associated with self-regulation. Though both tasks and surveys putatively measure self-regulation, they show little empirical relationship. Within tasks and surveys, however, the ontology identifies reliable individual traits and reveals opportunities for theoretic synthesis. We then evaluate predictive power of the psychological measurements and find that while surveys modestly and heterogeneously predict real-world outcomes, tasks largely do not. We conclude that self-regulation lacks coherence as a construct, and that data-driven ontologies lay the groundwork for a cumulative psychological science.

[1] Department of Psychology, Stanford University, Stanford, CA 94305, USA. [2] Department of Psychology, Arizona State University, Tempe, AZ 85281, USA. [3] Department of Psychiatry, Geisel School of Medicine at Dartmouth, Dartmouth College, Lebanon, NH 03766, USA. Correspondence and requests for materials should be addressed to I.W.E. (email: ieisenbe@stanford.edu)

Science is meant to be cumulative, but both methodological and conceptual problems have impeded cumulative progress in psychology. While a flurry of recent work has focused on the poor reproducibility of psychological findings[1], a more fundamental conceptual challenge arises from the lack of integrative theory development and testing. As pointed out by Newell[2] and Meehl[3] decades ago, psychological findings are rarely situated within the broader literature and the resulting theories are siloed and overspecialized. Thus it seems essential to develop an integrative framework, one that capitalizes on the wealth of psychological phenomena already described, to create a foundation for future inquiry[4,5]. We propose that the data-driven development of ontologies—formal descriptions of concepts in a domain and their relationships[6]—can serve as such a framework. By specifying psychological constructs and their relationship to observable measures, ontologies can serve as a lingua franca across disciplines, identify theoretical gaps, and clarify research programs[7]. In this paper, we integrate a large array of psychological measures into an ontological framework via a large-scale study of behavioral individual differences.

To begin this enterprise, we focus on the psychological construct of self-regulation, which refers to the ability to regulate behavior in service of longer-term goals. This domain is an ideal case study for ontological revision due to the substantial theoretical and methodological diversity associated with the construct[8–10] and its putative connection to a number of important real-world outcomes[11]. While ontological revision has been a central focus of recent work in self-regulation[10,12], it has yet to be tackled through quantitative modeling of data that span the domain.

Theoretic integration and construct validity[13] should be complemented by ecological validity. The constructs studied by psychologists are hypothesized to serve as building blocks for everyday behavior, and their dysfunction is thought to be central to many mental health disorders[14]. This is particularly true of self-regulation, which is putatively connected to a range of significant outcomes, including academic performance, health outcomes, and economic well-being[11,15,16]. However, psychological constructs are often derived to explain behavior in an ad hoc manner, rather than to generate a priori predictions of real-world outcomes, leaving ecological relevance largely untested[17]. Even when associations between psychological constructs and real-world outcomes are examined they rarely are evaluated using modern assessments of predictive accuracy (e.g., employing cross-validation), generally inflating estimates of predictive power[18]. We evaluate the ability of psychological measurements to predict a range of self-reported real-world outcomes, and unpack the predictive success in terms of the ontology. Linking disparate real-world outcomes based on ontological similarity is a critical step towards creating a generalizable science of human behavior.

We formalize psychological ontologies in terms of a quantitative psychological space in which psychological measurements are embedded, and a set of clusters that organize those measurements within the space. The space defines parametric features of mental processes while the clusters label a set of measurements that are close in the quantitative space, thus having a similar ontological fingerprint.

The first step of ontology discovery—defining a quantitative space—connects to a classic approach in psychology, factor analysis, which has been used to infer the dimensional structure of broad constructs such as personality[19,20] and emotion[21], inform measurement design[22], and drive hypothesis development[23]. Although many prior studies were limited by a small number of measures, recent work has expanded the size of the measurement batteries employed[24–26]. However, while the measurement number has increased, the diversity of measurements is often heavily circumscribed by a precommitment to the scope of a construct. Though dense measurement of a construct may be useful, this approach tacitly reifies the theoretic framing without testing it. For example, if one unpacks the multidimensional structure of impulsivity by only assessing putatively related measures, there is little chance the concept of impulsivity itself will be questioned. Occasionally, constructs are challenged on the basis of failing a test of convergent validity, but without a direct comparison between within-construct and across-construct correlations, the same magnitude of correlation can be used to both defend[9] and reject[27] the convergent validity of a construct. To create a holistic, quantitative space, it is thus necessary to widen the scope of the behavioral measurements analyzed.

We selected a set of 22 self-report surveys and 37 behavioral tasks (see Supplementary Methods; Supplementary Tables 1, 2) in order to capture constructs associated with self-regulation (e.g., impulsivity) while also including a set of measures reflecting diverse psychological functions that extends beyond those normally studied in the context of self-regulation (e.g., personality). This choice affords the possibility of identifying the borders of self-regulation as a construct or rejecting the discriminant validity of self-regulation itself. Once selected, behavior on each of these 59 measures was decomposed into multiple dependent variables (DVs; $n = 193$; 129 task DVs) which reflect means of specific item sets, comparisons between task conditions, or model parameters thought to capture psychological constructs (Fig. 1a, b). 522 participants completed this measurement battery using Amazon Mechanical Turk. A subgroup of 150 participants completed a retest on the entire battery, allowing estimation of retest reliability[28].

Using these DVs, we find that behavioral task and survey measures are largely unrelated. Informed by this divergence, we construct a psychological ontology composed of two low-dimensional spaces separately capturing task and survey DVs. The dimensions of these spaces highlight relational structure amongst the constituent DVs, and define reliable individual differences. Using individual differences scores derived from the ontology, we show that behavioral tasks fail to predict substantial variance in real-world outcomes thought to relate to self-regulation, while surveys perform moderately well. However, the predictive success of surveys is found to result from a heterogeneity of psychological constructs for different outcomes. Combined with the lack of relationship between tasks and surveys, these results argue against a coherent and general self-regulatory construct.

## Results

**Creating a psychological space.** Our first goal was to create a psychological space: a structure that quantifies distance between DVs, and provides a vocabulary to describe disparate behavioral measurements. A foundational question is whether surveys and task DVs should be captured within a single space. Because the battery included both surveys and tasks putatively related to the same psychological constructs (e.g., impulsivity), one would predict significant relationships between the two sets of DVs, supporting a joint psychological space (though weak relationships between tasks and surveys have been reported before, see refs. 9,26,29,30).

To address this goal, we evaluated the association between task and survey DVs. Neither measurement category could predict DVs from the other category, and correlations between measurement categories were weak (Supplementary Fig. 1). Visualizing the relationships between DVs using a graph demonstrates the independent clustering of the two measurement categories (Fig. 2). The low correlations between these two groups of measures suggest a top-level ontological distinction between the

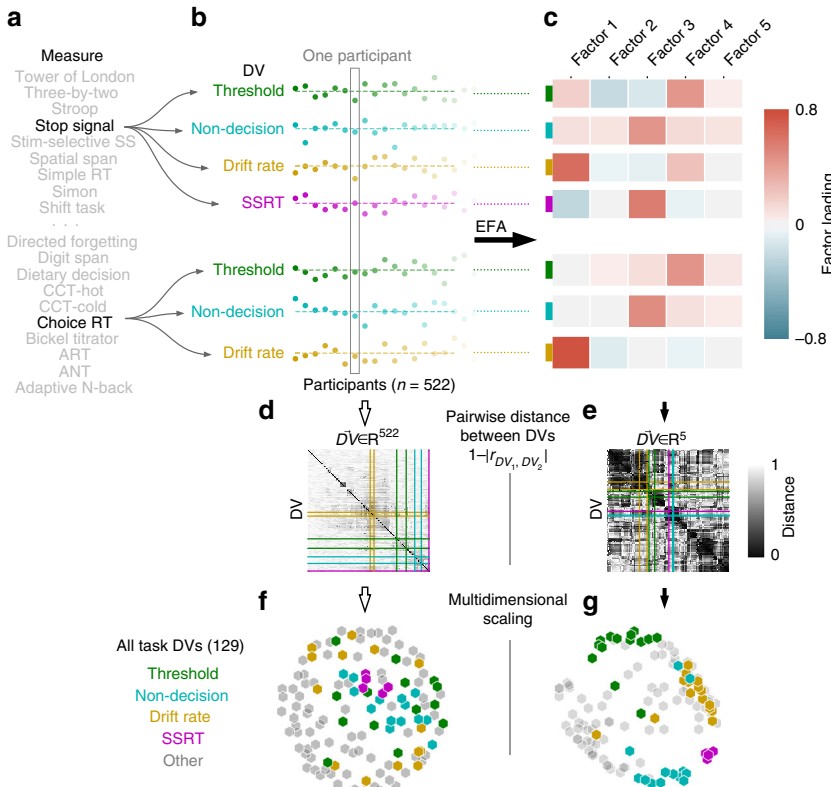

**Fig. 1** Summary of Task Analytic Pipeline. **a** Participants completed 37 separate task measures, of which a subset are shown. **b** 1st-level analysis of each measure resulted in a number of DVs. Choice Reaction Time and Stop Signal are shown as two example measures, from which 7 DVs are extracted. Participant scores are displayed as deviations from the mean for each of the 7 DVs. A subset of the 522 total participants are shown as individual dots. **c** EFA projects each DV from a 522-dimensional participant feature space to a lower-dimensional factor feature space. **d**, **e** Pairwise-distance between all 129 task DVs are shown for the participant space (**d**) and factor space (**e**). The variables from (**b**) are indicated by colored lines. **f**, **g** DV clusters are revealed in the lower-dimensional EFA space. Multidimensional scaling of the pairwise-distances in EFA space (**g**, see Methods) highlight obvious clustering, in contrast the participant space (**f**). DVs are colored based on type for visualization purposes only—actual analysis is wholly unsupervised

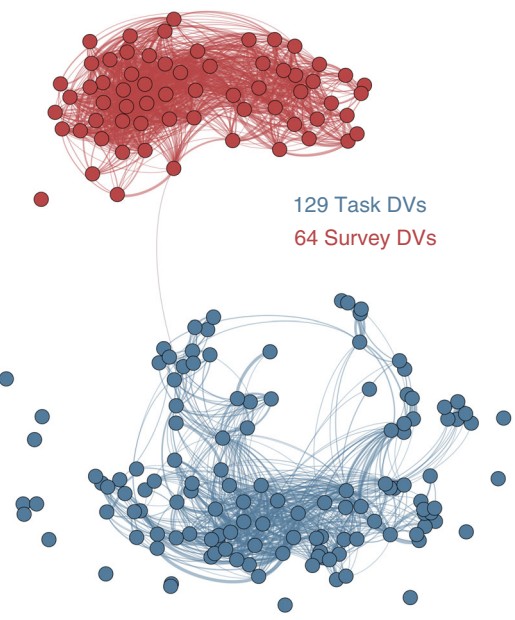

**Fig. 2** Relationships of all DVs. Graphical lasso[58] was used to estimate a sparse undirected graph representing the relationships amongst all DVs. Nodes represent the 193 DVs, while edges represent the estimated partial correlation between two DVs. DVs are colored according to measurement category (task DVs: blue) and edges have been thresholded (partial correlation strength >= 0.05)

constructs underlying task and survey DVs. We thus proceeded by constructing two psychological spaces. Future reconciliation of these two spaces may be possible, but would require the addition of spanning DVs that correlate with both task and survey DVs.

We defined task and the survey psychological spaces using exploratory factor analysis (EFA; Fig. 1c; Methods). An important step in factor analysis is dimensionality estimation. Using model selection based on the Bayesian information criterion (BIC), we found that 12 and 5 factors were the optimal dimensionalities (Supplementary Fig. 2) for the decomposition of surveys (Fig. 3) and tasks (Fig. 4), respectively. The robustness of the factor models was assessed using two methods: confidence intervals for factor loadings were created by bootstrapping, and EFA was rerun dropping out each individual measure (and all constituent DVs) to assess convergence of the factor solutions. Factor loadings were robust across bootstraps. Individual measures did affect the overall structure of the EFA models, particularly for the survey model, likely due to sparse measurement of highly discriminant psychological constructs (e.g., the survey factor Agreeableness was dependent on the inclusion of a Big-5 personality survey). See Supplementary Discussion and the online Jupyter Notebook (described in Methods) for a full description of the robustness analyses.

The survey EFA model fit the raw DVs better than the task EFA model (Survey $R^2 = 0.58$, Task $R^2 = 0.23$), but this difference is reduced once test-retest reliability of individual DVs is accounted for using attenuation correction[31] (Supplementary Fig. 3; adjusted survey $R^2 = 0.86$, adjusted task $R^2 = 0.68$).

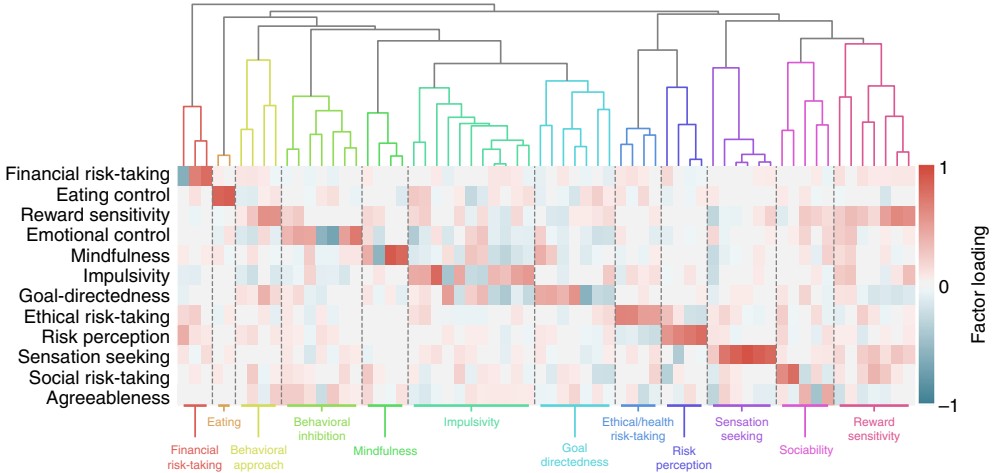

**Fig. 3** Survey ontology. 64 survey DVs were projected onto 12 factors discovered using EFA, represented by the heatmap. Rows are factors and columns are separate DVs ordered based on the dendrogram above. The dendrogram was created using hierarchical clustering, and separated into 12 clusters using DynamicTreeCut[34]. Identified clusters are separately colored, divided by dashed lines in the heatmap, and labeled at the bottom

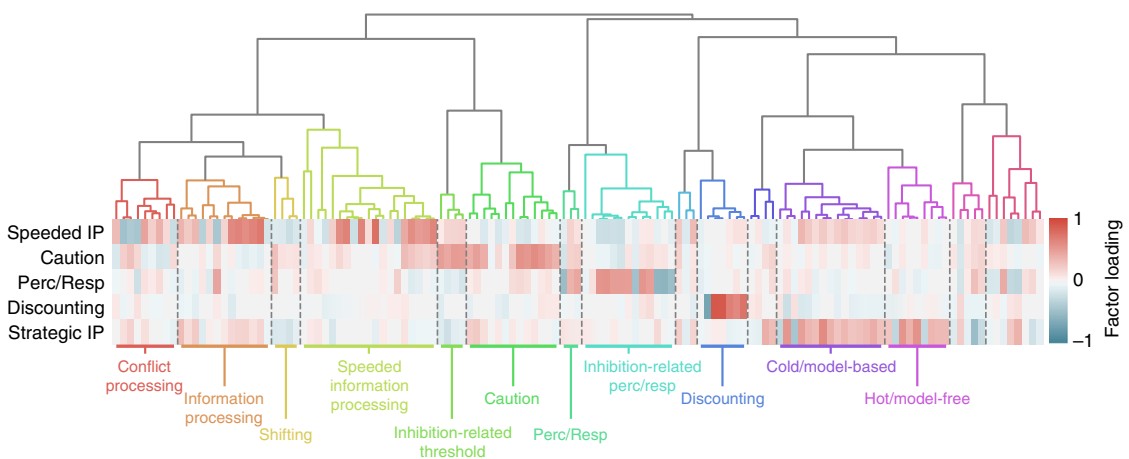

**Fig. 4** Task ontology. Identical to Fig. 3, except operating over 129 task DVs, which are projected onto 5 factors and separated into 15 clusters. IP: information processing

Interestingly, the factor scores for both tasks and surveys demonstrated high reliability (Fig. 5a, c) when evaluated in the retest sample, which equaled (for surveys) and exceeded (for tasks) the reliability of the constituent DVs (Supplementary Fig. 4). Additionally, it is evident that the variability within-participants across time was less than across-participant variability (Fig. 5b, d). Stability over time, and sensitivity to individual variability are central features of useful trait measures[32], and support the use of factor scores as individual difference metrics.

To understand the nature of the factors we evaluated the DVs that strongly loaded on each factor (these loadings are displayed in the online Jupyter Notebook). Most survey factors reflected separate measurement scales (e.g., Social Risk Taking and Financial Risk Taking derived from the DOSPERT) or a combination of several closely related DVs (e.g., sensation seeking, which related to DVs derived from the Sensation Seeking Scale, UPPS-P, I7, and DOSPERT). Thus the survey model recapitulates theoretical constructs in the field, albeit with a reduced dimensionality relative to the initial 64 measures, suggesting that survey measures are partially overlapping in the psychological constructs they represent. Notable exceptions to

this general recapitulation was the Goal-Directedness factor, which integrates a heterogeneous set of DVs related to goal-setting, self-control, future time-perspective, and grit, the Emotional Control factor, which relates to neuroticism, emotional control, behavioral inhibition, and mindfulness, and the Eating Control factor, whose selective association with emotional and uncontrolled eating supports its discriminant validity, which heretofore had not been assessed.

The task EFA solution resulted in 5 factors. The simplest factor was selective for temporal discounting DVs, which in turn only loaded on this factor. This result implies that temporal discounting tasks are largely divorced from all other task DVs, and uniquely probe a separable psychological function.

Three other task factors reflected components of the drift-diffusion modeling framework (DDM; see Supplementary Methods); Speeded Information Processing, Caution, and Perception/ Response were strongly and differentially related to drift rate, threshold, and non-decision time estimates, respectively. This supports prior findings[33] that individual differences in DDM parameters are correlated across a range of speeded RT tasks. While consistent with the DDM parameterization of decision-making

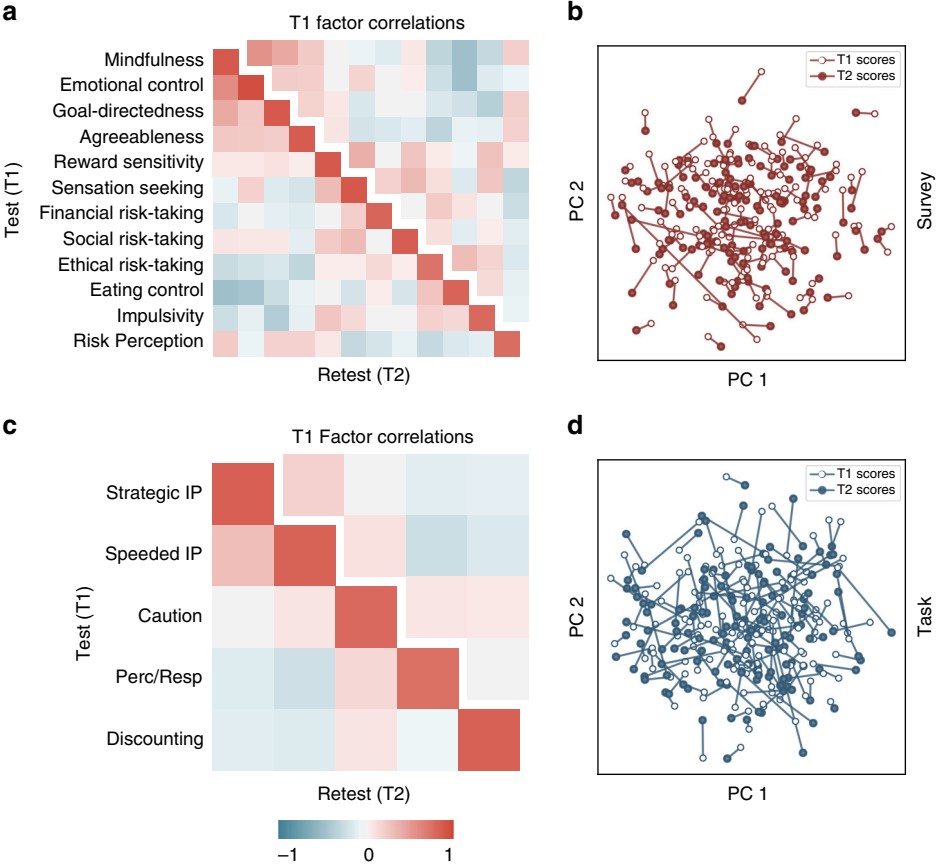

**Fig. 5** Retest reliability and correlations of factor scores. The EFA model derived from the full dataset ($n = 522$) was applied to a subsample of participants who repeated the entire battery within 4 months ($n = 150$) to compute factor scores (T2 scores). These factor scores were then correlated with factors scores derived from the same participants during their original testing (T1 scores). The lower triangle and diagonal of the heatmaps reflect these correlations and are displayed for both surveys (**a**) and tasks (**c**). The diagonal shows high factor score reliability within-individual across time. The separated upper triangle depicts the factor correlation structure at T1. To visualize the relative stability of individual factor scores compared to group variability, the factor scores were projected into a 2-dimensional space defined using principal components analysis (**b, d**). T1 scores are depicted using an empty circle and T2 scores are depicted using a filled circle, with each individual corresponding to one stick

processes, DDM measures were not exclusively associated with these factors, as other related DVs loaded sensibly (e.g., Go-NoGo d′ loaded on the Speeded Information Processing factor).

Finally, the Strategic Information Processing factor loaded on diverse DVs that were putatively related to working-memory, general intelligence, risk-taking, introspection, and information processing: generally tasks that were amenable to higher-order strategies, and unfolded on a time-scale greater than the speeded decision-making tasks modeled with the DDM.

It is worth noting that factors in both models (e.g., the Speeded Information Processing and Strategic Information Processing task factors) were moderately correlated (Fig. 5), implying a hierarchical organization.

**Clusters within the psychological space**. As a whole, both task and survey factors outlined sensible psychological dimensions that relate to many concepts discussed in the field. Given this, one might ask why other plausible constructs like self-control or working memory did not result in their own factors. However, because factors should be viewed as basis vectors for a psychological space, the principal concern is the subspace spanned by those factors, which determines the fidelity and generalizability of the DV embedding, rather than the specific direction of each

factor. The particular factors are ultimately a result of rotation schemes whose goal is interpretability—a useful objective to be sure, but one potentially divorced from the span of the psychological space. A consequence is that certain psychological constructs may emerge as clusters of DVs, rather than axes, in this space.

To identify clusters, we performed hierarchical clustering on the factor loadings of the DVs. Using this analysis, DVs that partially load on similar factors are clustered together. Hierarchical clustering creates a relational tree that affords clustering at multiple resolutions, depending on how the tree is cut. To identify theory-agnostic clusters we used the Dynamic Tree Cut algorithm[34], which accommodates different tree structures better than simpler methods that cut the tree at a fixed height. Doing so identifies 12 clusters for the survey DVs (Fig. 3) and 15 clusters for the task DVs (Fig. 4). Due to the nature of the trees' structure, clusters are hierarchically organized based on their relative distances in the psychological space. Alternative clusterings were compared using silhouette analysis (Supplementary Fig. 5), and clustering robustness were assessed using simulation and consensus clustering (see Methods). This latter analysis shows that clustering was moderately sensitive to small changes in factor structure, with some DVs more robustly clustered than others. As such, the clusters reported here should be taken with caution.

The relational trees and clusters are depicted as dendrograms in Figs. 4 and 5 for surveys and tasks, respectively. Details on all extracted clusters, including their constituent DVs and ontological fingerprint can be found in the online Jupyter Notebook. Here, we summarize the psychological content of the clusters.

The survey clusters largely mirror the dimensions of the psychological space. This implies that survey DVs generally interrogate a single psychological dimension, likely a consequence of research traditions that emphasize discriminant validity. A notable exception is the emergence of a self-control branch composed of two separate clusters: one primarily related to impulsivity (but also reflecting goal-directedness, mindfulness, and reward sensitivity), and one reflecting long-term goal attitudes, incorporating time-perspective, grit, and implicit theories of willpower. The compact, sensible nature of this branch suggests that self-control is a reasonable higher-order construct, at least in its ability to account for a set of psychological measurements.

In contrast to the simple relationship between survey dimensions and clusters, the task clusters capitalize on the full fingerprint of each DV to provide a complementary perspective of task structure. While a discounting cluster had a one-to-one correspondence with its corresponding dimension, the majority subdivide DVs that principally load on one dimension based on their secondary components. For example, the DDM threshold parameter governing choice on go-trials in stop-signal tasks clustered together (Inhibition-related Threshold), and separate from the threshold parameter in other speeded RT tasks (Caution). Given that the go trials on stop-signal tasks are analogous to other choice tasks, excepting the task context, this division indicates that people develop separate response strategies (e.g., preferring accuracy over speed) in inhibitory and non-inhibitory contexts. A similar finding can be seen within the DVs that load on Speeded Information Processing. The DVs that relate to conflict, broadly defined (e.g., the Stroop effect borne out in drift differences), separate from other Speeded Information Processing clusters. Non-Decision time similarly subdivides.

Towards the right of the task dendrogram are two clusters that most strongly load on Strategic Information Processing. These clusters reflect decision-making strategies, which have been previously described by dichotomies like "cold" vs. "hot" (though these terms are normally related to risk-taking[35]) or model-based vs. model-free[36]. There is also a working-memory (WM) component running through both clusters, with verbal WM tasks (digit span, keep track), associated with the model-based cluster, distinguished from the spatial span.

Four task clusters were less clearly related to existing theories (unlabeled in Fig. 4), containing sets of variables too ambiguous to be named. Each of these clusters contained relatively few DVs, thus requiring future work to disambiguate whether these clusters largely reflect noise or the beginning of sensible structure.

**Prediction of real-world outcomes**. Explicitly linking diverse literature is important for cumulative progress in psychology, but is not sufficient. Meaningful connection to real-world outcomes is also necessary to evaluate the generalizability of psychological theories. Although evaluation of this connection has historically been an important component of psychological research in the form of criterion validity, ambiguity regarding outcome measures, researcher degrees of freedom, publication bias, and inadequate tests of predictive ability limit our knowledge of how psychological measures relate to real-world behaviors[17,37].

To evaluate predictive ability, we used a broad set of self-reported outcome measures, including socioeconomic outcomes,

drug and alcohol use, and physical and mental health. We used EFA to reduce the dimensionality of the outcomes, creating 8 target factor scores (referred hereafter as "target outcomes") for each participant (see Methods and Supplementary Methods). Out-of-sample prediction was performed using cross-validation with L2-regularized linear regression to predict targets using factor scores derived from the task and survey EFA solutions (as well as other methods, see Supplementary Table 3). We created three separate predictive feature matrices: the 12 survey factor scores (Fig. 6), the 5 task factor scores (Fig. 7), and the combination of all 17 factor scores (Supplementary Fig. 6). We used factor scores rather than raw DVs due to their higher reliability, and to contextualize the predictive models within the psychological ontology. All analyses were repeated using the raw DVs themselves as predictors, which did not change the overall interpretation (Supplementary Table 4). We also performed the same analyses without cross-validation, which estimates the degree of over-optimism of in-sample associations.

Surveys exhibited moderate predictive performance, significantly predicting all target outcomes (randomization test: $p < 0.05$), with an average predictive $R^2 = 0.1$ (min: 0.03, max: 0.29; see Fig. 6). We visualized the standardized β coefficients of the predictive models to create an ontological fingerprint representing the contribution of various psychological constructs to the final predictive model for a particular target outcome (Figs. 6 and 8). Mental Health has a simple ontological fingerprint: Emotional Control alone was sufficient for prediction. Other fingerprints are more complicated, pointing to the contribution of multiple psychological constructs to these behaviors. For instance, Binge Drinking related to a combination of Risk Perception, Reward Sensitivity, Social Risk-Taking, and Ethical Risk-Taking. The fingerprints can also be inverted, giving a sense of which kinds of target outcomes are related to a particular psychological construct (Fig. 8). Doing so illustrates that while some constructs relate to only a few targets (e.g., the Emotional Control only predicts Mental Health), others, like Social Risk-Taking, are related to a range of targets.

In contrast to the surveys, tasks had very weak predictive ability (average $R^2 = 0.01$, max $R^2 = 0.06$, Fig. 7). While four target outcomes were significantly predicted above chance (p < 0.05), the mean $R^2$ for these four relationships was only 0.03 (max = 0.06).

The combined task and survey predictive model did not qualitatively differ from the survey predictive model, except when predicting Income/Life Milestones, where it performed better (Supplementary Fig. 6; Supplementary Table 3). The degree of overfitting when relying on in-sample prediction was modest when using the L2-regularized regression with the relatively small number of factor scores as predictors, but became a significant issue when the number of predictors increased (i.e., when using all DVs rather than factors). This analysis was also repeated using other prediction approaches, see Supplementary Table 4 for the results.

**Discussion**
The ontological framework provides insight into mental structure by synthesizing a multifaceted behavioral dataset. Of particular note is the lack of alignment of the same putative constructs across measurement categories and the low predictive ability of behavioral tasks. The former has precedent in the literature in a number of domains[24,26,30,38,39], which this work expands upon, suggesting that the inappropriate overloading of psychological terms (jingle fallacies) is widespread. The latter shows that psychological constructs associated with tasks lack substantial real-world relevance, and points to a need for greater emphasis of

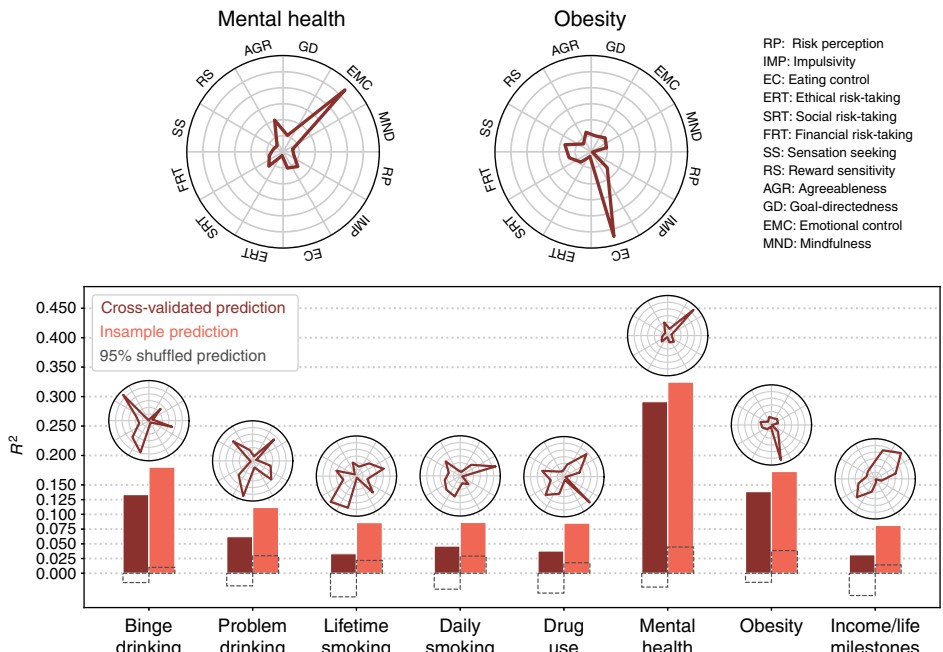

**Fig. 6** Prediction of target outcomes using survey factor scores. Cross-validated (dark bars) and insample (light bars) $R^2$ are shown. Dashed gray boxes indicate 95% of null distribution, estimated from 2500 shuffles of the target outcome. Ontological fingerprints displayed as polar plots indicate the standardized β for each survey factor predicted above chance (randomization test: $p < 0.05$). The ontological fingerprint for the two best-predicted outcomes are reproduced at the top. The y-axes are scaled for each fingerprint to highlight the distribution of associations—no inference can be drawn comparing individual factor magnitudes across outcomes (see Fig. 8 to do this)

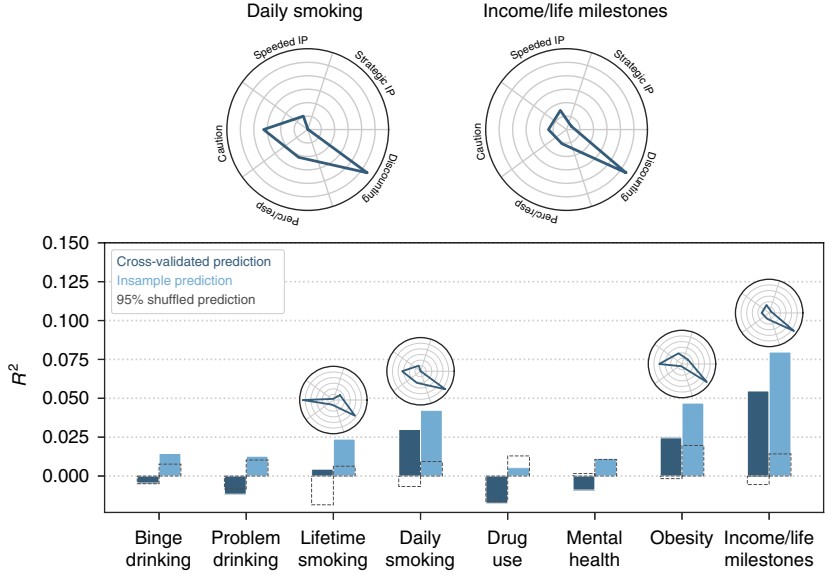

**Fig. 7** Prediction of target outcomes using task factor scores. Identical to Fig. 6, except for the truncated y-axis; the task factors are substantially worse at explaining variance in the target outcomes. IP: information processing.

predictive validity[17]. Together, these findings support the present pursuit of a revised psychological ontology.

This approach to ontology discovery fundamentally rests on correlations. Relationships amongst behavioral outputs define a structure connecting observable measurements to psychological constructs. Combined with explicit connection to real-world outcomes, the ontology provides a quantified structure reminiscent of a nomological network[13]. Furthermore, as the ontology is a function of measurement correlations, its structure is

immediately relevant for the many psychological hypotheses that are fundamentally about relationships amongst behaviors (e.g., the existence of overarching psychological traits like a general factor of intelligence[40], or risk preference[26]). For example, higher-order claims about the separability of various decision-making processing stages (in line with the DDM), and the discriminant validity of concepts like sensation-seeking are recapitulated by our factors. At the level of clusters, lower-order constructs are distinguishable (e.g., conflict-related information processing).

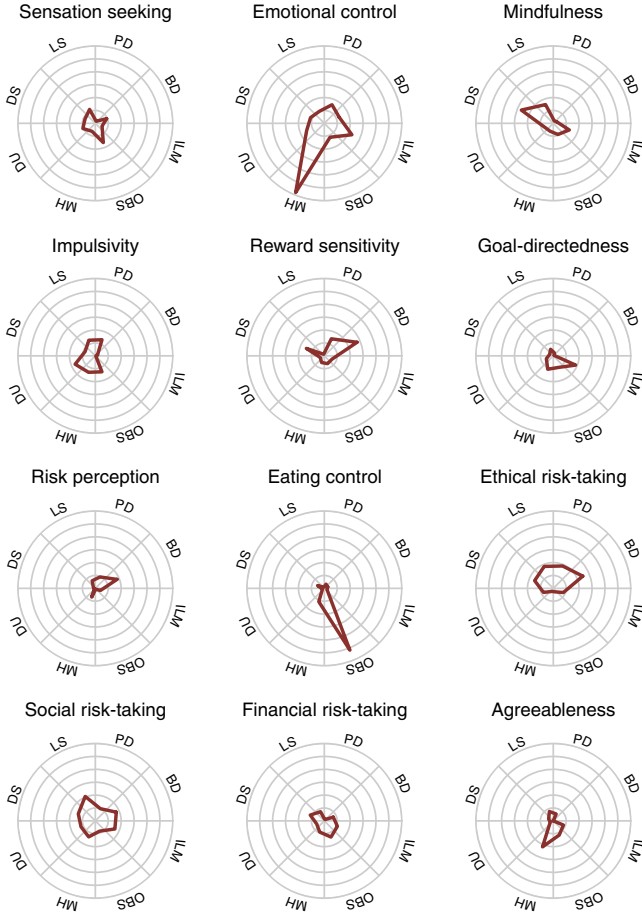

**Fig. 8** Survey factor ontological fingerprints. Each polar plot represents the relationship of each survey factor with the 8 target outcomes. This relationship is defined by the β value associated with that factor predicting that target outcome. For instance, the factor Eating Control is selectively predictive of obesity, and no other outcome. Unlike Figs. 6 and 7, the fingerprints all share the same y-axis (max value = 0.4) and thus can be compared across plots. For example, Impulsivity and Social Risk-Taking both contribute to the prediction of lifetime smoking, but the association with Social Risk-Taking is stronger. BD: binge drinking, PD: problem drinking, DU: drug use, LS: lifetime smoking, DS: daily smoking, MH: mental health, OBS: obesity, ILM: income/life milestones

At a finer detail, theory advancement concerning particular measures is also possible. For instance, previous work has shown that individuals titrate their reliance on model-free strategies in task contexts that strain working memory (WM) capacity[41]. In the present data, WM is negatively correlated with model-free decision making, which is borne out in the ontology as opposite loadings on the Strategic Information Processing factor. This implies that the tradeoff between WM and model-free decision making not only depends on task context, but is also modulated by individual traits—those with a larger WM span are more prone to model-based over model-free strategies. As a second example, the angling-risk-task (ART) is often interpreted as a risk-taking measure. However, some work suggests sequential risk-taking tasks like the ART instead measure an agent's ability to assess environmental statistics and act optimally, rather than a propensity towards risky action[26,42]. Our data support this latter view, as ART DVs cluster with working memory, decision-making and "hot" cognition DVs, and are unrelated to self-report measures of risk-taking (e.g., DOSPERT).

The breadth of the dataset underlying data-driven ontology development is also important. As an example case, stop-signal reaction-time (SSRT) DVs, putatively related to response inhibition, load on the same factor as non-decision time estimates, DDM DVs intended to capture perceptual and response processes. This suggests a relationship between these normally separable constructs. It is also apparent that without including both non-decision times and SSRT in the same measurement battery, a robust SSRT factor would likely be found and interpreted as response inhibition—thus an opportunity to bridge literature would have been overlooked. This same reasoning can inform the observed bifurcation of surveys and tasks. The absence of a spanning construct (e.g., one that relates to both behavioral measures and self-report surveys) prevents accurate estimation of the psychological distance between task and survey constructs. Serendipitously discovering such a construct will be difficult given the consistent findings that surveys and tasks are weakly related; instead, new research programs should make finding linking constructs their explicit objective.

Besides the particular measures we used, other relationships between constructs may be missed due to our method's reliance on correlation: it is only sensitive to linear bivariate relationships between DVs. While more complex unsupervised approaches may reveal non-linear relationships missed using our approach, they would likely require more data and be less interpretable. Interactions amongst three or more variables are also possible, but such analyses were not concordant with our directed exploration. We hope that the openly available dataset will allow others with specific hypotheses to test these more complicated interactive models.

This work is also limited by its convenience sample recruited via Mechanical Turk. As such, it may underestimate the full diversity of psychological functioning (e.g., variability associated with mental illness, or biases related to the Mechanical Turk population), and may not systematically measure the range of outcomes we are trying to predict (see below). Future studies using targeted sampling of specific populations with a similar battery could help address these issues.

In addition to serving as a bridge between research disciplines, the ontology also clarifies how psychological measurements relate to real-world behavior. The ontology defines individual traits whose retest reliability equals or surpasses the individual DVs. For tasks in particular, EFA integrates multiple noisy DVs and creates reliable measures of central psychological constructs. In doing so, EFA addresses a perennial critique of behavioral tasks: their poor psychometric properties limit their real-world applicability, particularly when it comes to predicting individual behavior[28,43,44]. However, though factor scores proved reliable, they demonstrated a weak relationship with the target outcomes.

Why did the surveys predict adequately, while the tasks did so poorly? The bifurcation of the ontology by measurement category suggests one explanation: tasks do not probe cognitive functions relevant for the target outcome measures. Such an explanation challenges current psychological theories of self-regulation, but allows for the possibility that the tasks would relate more strongly to other outcomes. An alternative, is that the contrived nature of behavioral tasks fundamentally compromises their ecological validity[45]. While the sensibility and reliability of the task factors speak to real structure in human behavior, psychology's reliance on controlled experiments may lead to theoretical overfitting. That is, theories that are explanatory and predictive of human behavior in experimental contexts may lack relevance for naturalistic human behavior. Expanding the scope of outcomes evaluated would aid in distinguishing these two explanations.

Regardless of the explanation, the task prediction results are in conflict with a narrative claiming strong relationships between

behavioral measurement and real-world behavior[46]. Part of this conflict stems from a difference in framing: we highlight the low variance explained, while other work focuses on statistical significance[47]. We believe this latter framing is somewhat misleading, and does not adequately reflect the poor state of prediction with behavioral tasks. That said, framing alone cannot fully explain this discrepancy. The widespread use of in-sample prediction (compared to cross-validation, employed in this work) undoubtedly also plays a role. Though in-sample prediction did not greatly inflate $R^2$ estimates here, studies with lower power would be more adversely affected. Because prediction using behavioral tasks is relatively weak, and often coupled with small sample sizes, the use of in-sample prediction plays a role in exaggerating the estimated predictive power of behavioral tasks. Coupled with other sources of false-positives (e.g., publication bias), it is likely that prediction work in psychology suffers from the same reproducibility issues that have plagued the field more generally[1].

In contrast, surveys predicted the target outcomes moderately well. This may be partially explained by methodological similarity, as both surveys and the real-world outcomes in this work are self-report measures that may be susceptible to similar biases[48]. More concretely, the relative predictive success says little about the causal direction. Survey measures may be partially dependent on a person's knowledge of their own behavior (e.g., "I drink heavily, therefore I am impulsive"), in a way that tasks are not. From this view, survey measures are merely a roundabout way to measure a person's real-world behaviors, rather than measurements of psychological constructs relevant for those behaviors. Fully addressing this criticism is beyond the scope of this paper, but its possibility, combined with the lack of relationship with behavioral tasks, challenges the construct validity of surveys as well.

Putting those criticisms aside, and assuming the general claim that surveys probe psychological traits, the ontological fingerprints imply that real-world outcomes rely on an overlapping mixture of psychological constructs. If these constructs are amenable to intervention, this framework supports the development of ontological interventions (e.g., aimed at reducing impulsivity) that cross-cut multiple real-world behaviors, similar in spirit to the taxonomy of self-control interventions proposed by Kotabe and Hofmann[10]. Particular behaviors like smoking could then be targeted with a multi-pronged strategy combining multiple ontological interventions. The converse is also true - the ontological fingerprints suggest which intervention targets are dead ends. Thus the ontology holds promise for a generalizable and cumulative science of behavior change.

Beyond quantifying individual variability, much of psychology is concerned with the interplay between individual traits and a person's environmental context[49,50] or state[51], which was not quantified in this study. One possibility, untested at scale, is that individual traits will relate more closely to real-world outcomes when the environmental context is properly accounted for. Modeling trait-environment interactions requires a compact and consistent description of environmental context, which would be enhanced by an ontological approach that articulates how to measure and integrate aspects of a person's environment into a quantitative whole[50]. Until that is developed, the present psychological ontology provides a concise vocabulary to define traits, allowing individual studies of trait-environment interactions to generalize their claims to a broader set of measurements and constructs.

Finally, this work provides suggestive evidence that psychology should move beyond the idea of self-regulation as a coherent construct. At minimum, the lack of convergent validity between surveys and tasks calls into question theoretical claims built on

assumed survey-task relationships. However, the challenge to self-regulation is greater than identifying this methodological bifurcation. Self-regulation is defined as the trait that putatively underlies an individual's capacity to achieve long-term goals. This implies a strong connection to real-world outcomes, a requirement that many of of the measures included in this study fail to meet. Although it's possible that environmental context must be evaluated to properly reveal a well-defined self-regulation construct, as mentioned above, a simpler explanation is that there is no unified trait, or set of traits, that engenders successful goal-attainment across contexts. Instead, self-regulation may be an emergent property—a label we ascribe to a suite of person-environment interactions that share little in common besides the general challenge of overcoming a desire-goal conflict[10]. While integrative theories of self-control[10] emphasize the importance of this conflict framing to link various self-control failures, the present work challenges such a view. We do not dispute the existence of systems that support constituent functions involved in self-control (such as delay discounting or conflict monitoring), but we see little evidence that it plays a large role shaping diverse life outcomes or health behaviors, independent of context.

With that said, the present ontology provides a means to conceptualize the suite of mental processes that may be relevant for how people successfully navigate the world. For example, a long research tradition on the unity and diversity of executive function (EF) has proposed a 3-factor model of EF composed of Inhibition, Updating, and Shifting[52]. While the particulars of the model are debatable[53], it has provided a theoretical framework for discussing EF, and an avenue towards a quantitative description of behavioral phenotypes[54,55]. Our approach, which situates these small number of EF measures within a larger array of measures capturing many mental processes, pushes forward the concept of unity and diversity beyond executive function to human behavior largely writ.

## Methods

**Extraction of individual difference measures**. Our dataset consisted of 522 adult participants, each completing a battery composed of 37 behavioral tasks and 22 self-report surveys. It included measures putatively related to self-regulation including risk-taking, temporal discounting and impulsivity, but also extended into more generic cognitive domains like working memory, information processing, learning, mindfulness, and others. By construction, some putative constructs like impulsivity were evaluated in both surveys and tasks, affording the opportunity to evaluate cross-measure consistency. In addition to these surveys and tasks, participants reported a number of real-world outcomes (e.g., self-reported questions relating to alcohol consumption, mental health, personal finances, etc.). Data were collected on Amazon Mechanical Turk using the Experiment Factory platform[56], which allows for easy replication of extension of this dataset. The study was approved by the Stanford Institutional Review Board (IRB-34926). All participants clicked to confirm their agreement with an informed consent form before beginning the battery. The data acquisition plan was pre-registered on the Open Science Framework (http://goo.gl/3eJuu1; though we deviated in several ways, see below). Additional information regarding data acquisition, the specific surveys, and tasks used, the selection of dependent variables, quality control, and data cleaning can be found in Supplementary Methods. We have also created Jupyter Notebooks to display figures and data that couldn't be contained within this article, available at this project's GitHub page: https://ianeisenberg.github.io/Self_Regulation_Ontology/.

**Deviations from pre-registration**. Our pre-registration described a graph-theoretic approach to ontology discovery. After implementing this approach, we recognized that it was imperfect for our purposes, and instead employed the combined factor analysis/clustering approach presented here. We decided that factor analysis would be more suitable for two main reasons: (1) factor analysis is a popular approach in psychology and thus is familiar to the field, and (2) factor analysis provides an ontological embedding for DVs, which combined with clustering provides two perspectives on the organization of psychological measures. Graph theoretic approaches are less common and would be restricted to identifying clusters without providing an embedding in an interpretable space.

We also deviated from the pre-registration by using a number of predictive models. Our pre-registration mentions random forests, which we include, but also make use of regularized linear regression models and SVMs. Initial analyses found

that random forests severely overfit the data, leading to our ultimate focus on Ridge Regression.

**Assessment of test-retest reliability.** The battery was notably divided by measurement type; self-report surveys and behavioral tasks. These measurement types potentially differed in their psychometric properties. Surveys are generally developed with psychometric theory in mind, and are routinely assessed for reliability. This is in stark contrast to tasks, where psychometric properties are often unknown, and rarely reevaluated.

We evaluated the reliability of all DVs in the same population by analyzing the subset of participants that completed the entire battery a second time. The full description of the subsequent analysis are laid out in Enkavi et al.[28]; summarizing, the surveys showed greater test-retest reliability (ICC3k $M = 0.80$, $SD = 0.06$) compared to the tasks (ICC $M = 0.45$, $SD = 0.21$, see Supplementary Fig. 4. There was substantial heterogeneity within tasks with some measures (e.g., discounting and DDM parameters) performing much more reliable than others. We used Pearson correlations between the two sessions as a measure of reliability, establishing a noise-ceiling (maximum predictive power possible given irreducible noise) to evaluate the fit of the exploratory factor analyses.

**Association between tasks and surveys.** Task and survey DVs had weak to no relationship with each other, as is evident by their uncorrected Pearson correlations (Supplementary Fig. 1a). To more rigorously quantify the relationship between tasks and surveys we employed two separate methods. First, we assessed how well a held out DV was predicted by either all task or survey DVs (excluding the to-be-predicted DV). This resulted in 4 distributions of predictions: two within-measurement predictions (task-by-tasks and survey-by-surveys) and two across-measurement predictions (task-by-surveys and survey-by-tasks). Prediction success was assessed by 10-fold cross-validated ridge regression using the RidgeCV function from scikit-learn with default parameters[57].

We also assessed this relationship by constructing a graph, where nodes are unique DVs and edges reflect the partial correlation between two DVs after conditioning on all other DVs. To estimate these correlations, we employed the Graphical Lasso[58] using the EBICglasso function from the QGraph package[59]. Visualization of the graph (Fig. 2) was accomplished using a force-directed algorithm in Gephi[60], with edges reflecting the absolute value of the partial correlations greater than 0.01. Results are described in Supplementary Methods and depicted in Supplementary Fig. 1.

**Exploratory factor analysis.** Exploratory factor analysis (EFA) seeks to explain the covariance of a number of observed variables in terms of a smaller number of latent (unobserved) variables, called factors. Each observed variable is modeled as a linear combination of these latent factors and some measurement error:

$$X - \mu = LF + \varepsilon \qquad (1)$$

where $X$ is an $m$ (DV) $\times n$ (participant) matrix of observed DVs, $\mu$ is a matrix of variable means, $L$ is the $m \times f$ (*number of factors*) loading matrix, describing the relationship between each variable and the latent factors, and $F$ is the $f \times n$ matrix of factor scores. $\varepsilon$ captures measurement error—the variance left unexplained by the latent (common) factors. In the current study each DV is represented by 522 participants—the individual participant scores—and EFA is used to estimate the embedding of these DVs (represented by the loading matrix) in a common psychological space spanned by the latent factors. Once estimated, factor scores are computed, representing the degree to which an individual represents that latent factor. For example, we used EFA to reduce the outcome measures to 8 factors, which are then used to compute 8-factor scores for each participant. These factor scores became the outcome targets. One outcome target related to variables related to binge drinking, and did not relate to any other variable—thus its related factor score represents an individual's general tendency to binge drink, and was named accordingly.

EFA was performed using maximum likelihood estimation, followed by oblimin rotation to rotate the factors without enforcing orthogonality. Factor rotation leads to easier interpretation by optimizing "very simple structure"[61], without changing the fit of the model. Factor scores were estimated using the tenBerge method, which is most appropriate given oblique rotation[62]. All analyses were implemented using the "fa" function from the psych package in R[63].

An important step when performing EFA is deciding on $f$, the number of factors to estimate. Though there are many procedures to accomplish this, we chose the number of factors that minimized the Bayesian Information Criteria (BIC). BIC is a criterion for model selection that attempts to correct for overfitting by penalizing more complex models, with lower values represent a better balance between capturing the data and model complexity. Supplementary Fig. 2 displays BIC values for EFA solutions with different numbers of factors. Other criteria identified an overlapping range of optimal dimensionalities, consistent with the notion that there is no single best dimensionality[64].

The optimal solutions for tasks, surveys and outcome variables all had cross-factor correlations after oblimin rotation. The correlations amongst factors are displayed in Fig. 5.

**Factor score robustness.** Confidence intervals on factor loadings were calculated using the "iter" option from "fa" function from the psych package in R, with the fraction of samples kept in each sample set to 90%[63]. This function runs EFA on 1000 bootstrapped samples, and uses the results to calculate the mean and standard deviation of loadings. The robustness of the factor models was also assessed by dropping out each individual measure and rerunning the analysis. Some measures had large effects on the discovered factors, while others were inconsequential. The results of these robustness analyses are communicated in the online Jupyter notebook and Supplementary Discussion.

**Factor analysis communality and DV test-retest reliability.** Communality refers to the variance accounted for in the DVs by the EFA model. Average communality (equivalent to overall variance explained by the EFA model) was greater for survey DVs ($M = 0.58$, $SD = 0.17$) than task DVs ($M = 0.23$, $SD = 0.21$), and differed between different DVs. Though this is partially explained by the different number of factors identified using the BIC criterion (5 factors for tasks, 12 factors for surveys), a 12-factor task model still only had an average communality of 33.

There are two main explanations for low communality: either the estimated factors do not span a psychological space that properly represents all DVs (e.g., the factors are a poor model for the data) or the DVs themselves have poor measurement characteristics. The latter creates a noise ceiling, and puts an upper bound on the variance that can be explained by any model.

To investigate this we correlated communality and test-retest reliability (as measured by Pearson correlation). We only evaluated DVs which had a test-retest reliability above 0.2. We found a strong correlation between communality and test-retest reliability in the tasks ($r = 0.53$), and a smaller correlation between communality and test-retest reliability in surveys ($r = 0.35$), suggesting that measurement characteristics are related to differential communality across DVs. We adjusted for test-retest reliability by dividing the communality values for individual DVs by their squared test-retest reliability, which results in an adjusted measure of variance explained (i.e., attenuation correction[31]). After adjustment the task factor model explained 68% of the explainable variance (across DVs $SD = 0.39$), while the survey factor model explained 86% (across DVs $SD = 0.24$) (Supplementary Fig. 3). Thus the discrepancy in explained variance can largely be understood in terms of the poor measurement properties of task DVs.

**Factor score reliability.** Though task DVs were less reliable than surveys in general, it was possible that factor scores derived from the task EFA model were just as reliable as the survey factor scores. The intuition is that by integrating over many noisy measurements of a central psychological construct, EFA creates a reliable individual trait, much as survey summary scores are more reliable than the specific items that constitute that scale.

To evaluate this we made use of the 150 participants who completed the entire battery a second time (see "Assessment of test-retest reliability"). Factor scores were computed at both time points making use of the weight matrix derived from EFA run on the first completion (i.e., the same linear combination of DVs was used to create factor scores at both time points). Reliability was quantified by the Pearson correlation between factor scores at both time points. All 5 task factors ($M = 0.82$, min = 0.76, max = 0.85) and 12 survey factors ($M = 0.86$, min = 0.75, max = 0.95) proved highly reliable (Fig. 5). We repeated this analysis using intraclass correlation[65], which did not qualitatively change the conclusions (ICC3k, task $M = 0.90$, min = 0.86, max = 0.92; survey $M = 0.92$, min = 0.86, max = 0.98). This analysis was repeated using weights derived from EFA run on only the 372 participants who were not part of the retest cohort. Applying the independently derived weight matrix to the 150 retest cohorts at both time points did not change the reliability estimates (Pearson's $r$ task $M = 0.81$, min = 0.77, max = 0.84; survey $M = 0.86$, min = 0.77 max = 0.95).

To visualize the multivariate stability of the factor scores we projected the 5 (task) and 12 (survey) dimensional scores from both time points into two dimensions using principal components analysis. The two dimensions captured 52% of the variance of task factor scores, and 41% of the variance of survey factor scores.

**Hierarchical clustering.** Hierarchical clustering is a family of algorithms that builds a relational tree. We used an agglomerative clustering technique that iteratively combines DVs (separately for surveys and tasks). This technique relies on a predefined distance metric, which defines how clusters should be combined. We used correlation distance (or dissimilarity) as our distance metric. Because of the arbitrary direction of our measures (e.g., an "impulsivity" DV could easily be represented by a flipped "self-control" DV) we used absolute correlation distance, defined as:

$$distance = 1 - |r| \qquad (2)$$

We did not compute the correlation distance in native (participant) space (forgoing dimensionality reduction), but rather in the factor analytic embedding space defined by the loading matrix. We clustered DVs using the factor loadings rather than participant scores for two reasons: (1) clustering using factor loadings immediately situates each cluster within the interpretable psychological spaces defined above, and (2) projection into factor space more clearly separates DVs into

meaningful and discoverable clusters (Fig. 1d–g, Supplementary Fig. 5), suggesting that dimensionality reduction via EFA functions as a useful denoising step, and aids with the curse of dimensionality[66]. That said, following this initial analysis, we performed hierarchical clustering in native space for both surveys and tasks. The silhouette analysis (see "Clustering comparison and robustness", below) in Supplementary Fig. 5 shows that this clustering is worse, which was confirmed visually using similar dendrogram plots as used in the main text.

The hierarchy created by this technique has no intrinsic cut points, and thus no objective clusters. To identify clusters which are interpretively useful, we used the Dynamic Hybrid Cut Algorithm from the DynamicTreeCut package[34]. In comparison to naive approaches, which cut the dendrogram at a particular height to identify clusters, the dynamic tree cut algorithm cuts the tree at different heights depending on the structure of the underlying branch. We separately evaluated clustering using a simpler partitioning algorithm—cutting the tree at a single height in order to maximize the mean silhouette score. At most heights, the silhouette score is comparable to the dynamicTreeCut clustering solution, except at very low cut heights, which produce many small, uninterpretable clusters (Supplementary Fig. 5). Finally, if we compare the clustering solution produced by dynamicTreeCut to a single height cut that produces the same number of clusters we find good convergence between the clustering solutions, as quantified by the adjusted mutual information score (AMI) between the two clustering solutions (task $AMI = 0.91$; survey $AMI = 0.88$).

**Clustering comparison and robustness**. One metric used to compare the quality of hierarchical clustering solutions was the average silhouette score for each solution using scikit-learn[57]. This score is first calculated for each DV, and is a function of the DVs mean intra-cluster distance and mean nearest-cluster distance, and is then averaged across all DVs. The score ranges from the worst value of $-1$ to the best value of 1, with values near 0 indicating overlapping clusters. Though we did not rely on this metric to select our clusters, it was used to corroborate the Dynamic Tree Cut algorithm, and supported our two-step process of clustering after dimensionality reduction. Silhouette scores for a range of different clustering approaches (before and after dimensionality reduction, using a fixed-height cut, and using Dynamic Tree Cut) are displayed in Supplementary Fig. 5.

The robustness of hierarchical clustering solution to data perturbations was also assessed. To do so we simulated 5000 separate loading matrices, where every element of each loading matrix was a sample from a gaussian whose mean and standard deviation were derived from the bootstrapped factor analysis (see "Factor score robustness"). We also dropped out 20% of the DVs randomly. Clustering analysis, identical to the main analysis, was then performed on these simulated loading matrices. The simulated clusters were compared to the original clustering using adjusted mutual information (AMI). AMI ranges from 0 to 1, with 1 indicating perfect agreement across clustering solutions and 0 indicating no overlap.

In addition, to evaluate the robustness of pairs of DVs clustering together we calculated the percentage of times DV pairs co-occurred within a particular cluster, for all DV pairs, across the simulations. This results in a co-occurrence matrix, which can be used as a distance matrix for consensus clustering. We compared the consensus clustering solution to the original clustering solution using AMI. Finally, we used the clusters reported in this work (Figs. 3 and 4) to calculate the average co-occurrence of DVs within-cluster, out-of-cluster, and between each DV and DVs in the two closest clusters. The results of these analyses are reported in the Supplemental Discussion and the online Jupyter Notebook.

**Visualization using multidimensional-scaling**. In Fig. 1, task DVs and a priori DV clusters (drift rate, threshold, non-decision time and stop-signal reaction time) were represented using multidimensional-scaling (MDS) for visual convenience. MDS is a conventional visualization technique that distills higher dimensional relationships into two dimensions and was performed using scikit-learn's MDS function[57]. This visualization is intended to highlight the utility in clustering after dimensional reduction, but is not directly relied upon for any subsequent analyses.

**Prediction analysis**. The primary prediction analysis used the factor scores from tasks or surveys, as well as both combined, as features to predict outcome targets. These outcome targets were derived from EFA using identical procedures to the surveys and tasks on the individual outcome items (e.g., household income, cigarette habits, see Supplementary Methods). Prior to running EFA, age and sex were first regressed out of each outcome variables using simple linear regression. This procedure yielded 8 factors: Binge Drinking, Problem Drinking, Unsafe Drinking, Drug Use, Lifetime Smoking, Daily Smoking, Obesity, Mental Health, and Income/Life Milestones (see Supplementary Fig. 7 for the factor score correlations and the online Jupyter notebook for the full factor loading matrix). We used two different regularized linear regression methods to perform prediction: lasso and ridge regression, which differ in the form of their regularization. We also used two nonlinear regression methods: random forest and support vector machines. All methods used scikit-learn[57].

Cross-validation was performed using a balanced 10-fold procedure (custom code based on ref.[67]), thus fitting each model with 469 participants and testing on 53 left out participants. Across all folds, each participant's outcome factor scores

(the prediction targets) were predicted in a cross-validated manner. These estimates were correlated with the actual outcome factor scores to compute $R^2$. Insample $R^2$ were estimated by fitting identical models as above to the whole dataset and testing on the same dataset. Mean absolute error (MAE) was computed analogously. Cross-validated and insample $R^2$ and MAE for each model are shown in Supplementary Table 3. Ridge and lasso regression performed comparably, while nonlinear methods, particular random forests, overfit the data producing poor fits. Ridge regression was used to assess feature importance due to its desirable regularization properties compared to lasso (sparse feature selection was not necessary for interpretability with so few predictors) and comparable performance. Feature importance for the ridge regression (as shown in the ontological fingerprint polar plots in Figs. 6, 8) is defined as the standardized β coefficients.

Prediction results combining task and survey factor scores did not differ qualitatively from the prediction results using survey factor scores alone, except for a slight improvement for obesity and income/life-outcomes, which were the two targets where tasks performed above chance (Supplementary Table 3). This constitutes weak evidence that, for some targets, tasks can complement surveys to create a predictive model for real-world behavior.

One potential issue with our prediction analysis is the possibility of data-bleeding between cross-validation folds as a result of the factor analytic models. That is, the EFA models for both predictors (e.g., survey factor scores) and targets (outcome target factor scores) were fit on the entire dataset. This data-bleeding could inappropriately inflate prediction estimates. To control for this possibility we created an empirical null distribution of prediction success by shuffling the target outcomes and repeating the prediction 2500 times. 95% prediction success is shown in all prediction plots and is used as a significance cut off ($p < 0.05$) to display ontological fingerprints.

Complementing our prediction using task and survey factor scores derived from EFA, we performed the same analyses using the individual DVs (separately for tasks and surveys) as predictor features. These results are discussed in the Supplementary Discussion.

**Reporting summary**. Further information on research design is available in the Nature Research Reporting Summary linked to this article.

## Data availability
The imputed data underlying the analyses in this work, as well as the task and survey loading matrices, can be found on OSF [https://mfr.osf.io/render?url=https://osf.io/4j9hd/?action=download%26mode=render]. Raw and processed behavioral data, including trial-by-trial data for each task, are available at [IanEisenberg/Self_Regulation_Ontology], [https://github.com/IanEisenberg/Self_Regulation_Ontology/tree/master/Data/Complete_02-16-2019].

## Code availability
The data cleaning procedures, and analysis code for ontology construction and predictive work are also available at [IanEisenberg/Self_Regulation_Ontology]. The experimental code and first-level analysis code used to derive measure dependent variables are part of the Experiment Factory[56].

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

## Acknowledgements

We thank V. Sochat for developing the Experiment Factory infrastructure, and J. Wright, K.J. Gorgolewski, D. Birman, and the SOBC network for discussions and suggestions. We thank the Texas Advanced Computing Center and the Stanford Research Computing Center for providing computational resources that contributed to this research. This work was supported by the National Institutes of Health (NIH) Science of Behavior Change Common Fund Program through an award administered by the National Institute for Drug Abuse (NIDA) (UH2DA041713; PIs: Marsch, LA & Poldrack, RA). Additional support was provided by NIDA P30DA029926.

## Author contributions

Conceptualization: I.W.E., P.G.B., A.Z.E., D.P.M., L.A.M., R.A.P.; Methodology: I.W.E. P. G.B., A.Z.E., D.P.M, R.A.P.; Software: I.W.E., A.Z.E., J.L.; Formal Analysis: I.W.E., A.Z.E., R.A.P.; Investigation: I.W.E., A.Z.E., J.L.; Resources: R.A.P.; Data Curation: I.W.E., P.G. B.; Writing: I.W.E., P.G.B., R.A.P., Visualization: I.W.E., Supervision: P.G.B., R.A.P.; Funding Acquisition: L.A.M., R.A.P.

## Additional information

**Competing interests:** The authors declare no competing interests.

**Journal Peer Review Information:** *Nature Communications* thanks Renato Frey and the other, anonymous, reviewer(s) for their contribution to the peer review of this work. Peer reviewer reports are available.

