## [Peer Review File · Nature Communications]

Reviewers' comments:

Reviewer #1 (Remarks to the Author):

This paper reports a large study of 522 participants (recruited online on Amazon mTurk) who completed 23 questionnaires and 37 tasks related to "self-regulation". The authors make a general claim regarding cumulative psychological science, arguing that data-driven ontologies lay the groundwork to uncover mental structures. In particular, the authors argue that past psychological research a) suffered from siloed scientific traditions and b) focused on explanation rather than prediction, and claim that their approach -- in which self-regulation is almost used as a "case study" to illustrate this general point -- can overcome these issues. Their "data-driven ontology discovery" approach identified a strong gap between tasks and surveys. Further, the authors conclude that within the two classes of measures the ontology reveals "opportunities for theoretic synthesis" and that there are stable individual traits. Finally, surveys turned out to generalize moderately to (self-reported) real-life outcomes, whereas tasks did not.

This really is an exciting project, and I am particularly impressed by the effort that the authors have invested in setting up the entire online framework to assemble this large dataset, as well as by the numerous detailed and (mostly) informative analyses. Moreover, in line with current open science standards, the data-collection plan was pre-registered (but not the exact analysis plan), and the materials, data, and analysis code are openly available such that everything can be reproduced. Therefore, this study definitely deserves publication. That said, I have several conceptual questions and comments regarding the framing and scope of the current manuscript.

Specifically, the authors make a strong claim regarding cumulative science and the need for data-driven "ontology creation". In principle I agree with these points, but such an approach also has its risks. In an ideal world, applying a purely data-driven approach to a sufficiently large number of diverse measures might indeed reveal a series of clear-cut cognitive dimensions. Of course, to be truly informative such dimensions would have to be further validated (e.g., with imaging methods, by demonstrating substantial incremental predictive validity for specific life outcomes, etc). However, if such a validation fails or provides only moderate evidence, theory development is naturally very limited. So does the present study succeed in "ontology creation"?

For several reasons mentioned below, the "ontology" of self-regulation remains quite opaque to me after having read this article, despite this impressive and large data-set, and despite the richness of all the analyses. Rather, the reader is left with several open questions (regarding the ontology of self-regulation itself but also regarding the usefulness of a purely data-driven approach). Given that this approach, in my view, did not quite live up to the expectations that the authors raised in the introduction, it is all the more a pity that there is no proper review of what is already known regarding (the ontology of) self-regulation and related constructs, which would have permitted contextualizing the present findings therein. This lack of integration into the current literature limits the impact of the present findings (which are informative and interesting even if the entire ontology remains somewhat unclear), which is somewhat paradoxical given the authors' strong claim for "cumulative science".

My general recommendation is therefore that the authors revise the article with a somewhat more "balanced" perspective, adopting a more critical view regarding how much we have actually learned about the ontology of self-regulation in this study, including a discussion of the risks of such a purely data-driven approach. Please find below a series of suggestions / and comments that I hope are useful to this end.

General comment:

* The authors have already published an article in "Behavior Research and Therapy" (Eisenberg et al., 2017), in the context of which (preliminary) analyses have been conducted using the same sample (but no results are reported there), and where they claim to focus on "ontology creation" (which is, as I understood, the main goal of the present article). Even though this article is cited once in the present article, there is absolutely no discussion of how the two articles relate to each other. My reading is that the original article functions as kind of a pre-registration. If this is true, the authors should explain which changes in the analytic process they implemented, and why they did so. Moreover, the original article mentions additional components of this study that are not explained here, at all (imaging, mobile assessments, etc.).

Major conceptual points regarding the ontology / ontology discovery:

* The authors hardly discuss the identified dimensions (i.e., the factors), even though they are supposed to be the core elements of an ontology (there are some further details in the supplemental materials and online, but the interpretation of these results is largely up to the reader). This is surprising and certainly does not help conveying what this ontology actually entails. Furthermore, it is somewhat unclear how strongly the inclusion of particular measures influences which dimensions are eventually identified. Consequently it remains somewhat vague how strongly the identified dimensions are indeed part of the "ontology of self-regulation", particularly given the absence of a strong rationale for the measurement selection (including the set of "broader measures").

* Of course, a complicating factor in this respect is that the identified dimensions do not span both measurement classes. However, by now it is well established that there are substantial gaps between tasks and surveys of various (highly related) psychological constructs, potentially due to purely measurement-related reasons (e.g., Harden et al., 2016; Frey et al., 2017, which has almost an identical analysis and plot). In principle, this strong empirical finding does therefore not rule out the possibility that there could still be a clear ontology of self-regulation. I did not find it overly plausible to assume that this ontology consequently entails two separate "psychological spaces".

* The transformation of the psychological spaces into clusters is an interesting approach. However, essentially nothing is done with these clusters here (the respective results section is rather a methods section, and only half a paragraph actually reports a few results, namely, that 13 clusters emerged, and a very brief interpretation of one branch and its two subbranches). It is left entirely open whether the ontology of self-regulation ultimately consists of the identified dimensions (factors) or rather the hierarchical clusters.

* Finally, whereas there was an analysis on cumulative R^2 's for the prediction of life outcomes, no analyses are reported regarding whether specific dimensions systematically relate to particular life outcomes, but not others. Furthermore, given that the factors were also correlated with each other, it remains unclear how "important" (and consequently, how robust) they actually are, and whether it is justified to consider them elements of the ontology of self-regulation.

In light of these points, what have we really learned regarding the ontology of self-regulation from this data-driven approach? I think a fair conclusion would be that this study provided a series of important incremental findings regarding self-regulation (which factors might underly this construct; how strongly do these dimensions cumulatively relate to life outcomes?), and corroborated the observation that tasks and surveys of psychological dimensions do not converge well (e.g., Lönnqvist et al., 2015; Harden et al., 2016; Frey et al., 2017). However, I do not think that the current approach really lives up to all the expectations the authors raise in the introduction. The findings would clearly be more informative were they better embedded in the existing literature (see next comment).

Integration with literature:

* The authors make a strong claim for cumulative science and criticize siloed research traditions. However, they themselves hardly review the theories and empirical findings on self-regulation: What is known regarding self-regulation and its overlap with related constructs? And how has this past knowledge been obtained? What psychometric / modeling approaches do exist in the literature? One can certainly criticize aspects of past research, but the authors could differentiate a bit better how their approach is really novel, and in what respects it actually implements elements that have already been used in the past.

* For instance, theories in the fields of personality or intelligence have been refined iteratively for about a century, using various measures and similar data-driven methods as implemented here. Moreover, there were assessments of the convergent and discriminant validity with other (potentially related) constructs and their predictive validity for real-life outcomes (e.g., White et al., 1994). It would be useful to discuss this research here.

* Relatedly, I see the main novelty of the present study in the large and diverse dataset of self-regulation measures -- as it has been done in recent attempts for related constructs. However, most of the methods implemented here are quite established in psychometrics, and the "data-driven approach" is in fact not as novel as the article implies. By largely neglecting this literature and using their own terminology, the authors raise the expectations that the current approach is fundamentally different from previous methods (see also previous comment). For example, they claim to do "data-driven ontology development" based on "statistical techniques that capitalize on similarity between variables", and that this strategy is "derived from a classical approach in psychology, factor analysis". Why not call a spade a spade and use the established terminology, rather than raising the impression that some novel and magic methods are going to be implemented here?

* The same is true for the term "ontology", which seems to be used almost exclusively by the authors of this research group. According to the authors, an ontology specifies latent psychological constructs (minor comment: this is a misnomer, as a "construct" is latent by definition; it should either read a "latent variable" or a "[psychological] construct"; Lilienfeld et al., 2015) and their relationship to specific measures. Please explain how "ontology" differs from concepts such as a "nomological network" or more generally, the "psychometric structure" of a construct -- both of which are established terms in the literature and refer to how / whether various (cognitive) processes involved in particular constructs relate to each other (e.g., Stahl et al., 2014). Somewhat ironically, the authors seem to fall prey to a "jingle-fallacy" (l. 258) themselves due to the idiosyncratic (and in my view unnecessary) use of such specific terms.

* Besides this idiosyncratic use, the term "ontology" seems completely underspecified: The authors use the term in all possible ways, such as: ontology discovery, ontology creation, ontology construction, ontology development, cognitive ontology, data-driven ontology, ontological framework, ontological factors, ontological similarity, ontology revision, ontological fingerprints, ontological interventions. As such, this term becomes pretty much meaningless. For example, does an ontology inherently exist and needs to be discovered, or does an ontology need to be constructed / developed? Does it refer to the psychological structure or is an ontology data-driven? Of course, the authors are free to use whichever terms they prefer, but this is certainly not facilitating theory integration / development (cf., cumulative science!).

Methodological comments:

* The present study clearly shines in terms of the number of implemented measures (60) and data-points per participant (196). However, in comparison the sample size is relatively small ($N = 522$), resulting in a subject-to-item ratio of 2.7:1 -- which seems very low for a data-driven exploratory approach. One (yet potentially outdated) rule-of-thumb has been to use subject-to-item ratios of no smaller than 40:1 (thus substantially larger), and simulation analyses have shown that already ratios of 20:1 lead to error rates "well above the field standard of .05" (Costello and Osborne, 2011). Is the extracted factor structure (i.e., "ontology") therefore indeed robust enough?

* There was a substantial variability in the time intervals for the retest-assessment. How was this taken into account in the analyses? Moreover, how were the bootstrapped ICCs exactly constructed (the reference is an unpublished ms. but the details should be provided here, too).

* The section on predictive validity starts with a criticism of what went wrong in past psychological research. While I agree with several points, I disagree that the methodology of the current study can overcome all of these issues and "allows for a generic evaluation of the state of behavioral prediction" (l. 209): On the one hand, there are of course also many researchers degrees of freedom here (e.g., selection of task and survey measures, selection of real-life measures, choices of how latent variables were extracted, etc.). For example, I could not find any information regarding the EFAs conducted for the real-life measures. Why were there exactly 9 factors? How independent were the different factors (in particular the different drinking and the different smoking factors might be highly correlated with each other). Factor inter-correlations (for tasks, surveys, and outcomes measures) should definitely be reported in the manuscript and not merely in the "online notebook" (with explicit numbers instead of the colored correlation tables). Furthermore, there is some obvious measurement invariance between the survey measures and the real-life measures, which are a) self-reported, too, and b) collected in the same assessment period (i.e., this is not really "prediction"). Therefore, the authors should be a bit more cautious here and not oversell this analysis.

Analyses / results:

* As a general comment, I found the various analyses very interesting and largely informative. However, as the exact analysis plan was not pre-registered and as there is no clear overview, some analyses in this paper (and the extended data section) feel quite exploratory. This could be improved with an overview of the analysis plan. Moreover, several analyses (in particular in the online materials) felt a bit like a "menu to choose from" ("we did it like this but also like that"), leaving the reader puzzled why each of these analyses is relevant.

* The following is an illustration of the previous comment: To visualize the "psychological space" the authors use both [partial] correlations between DVs (i.e., in the EVs, Figures 1 and S2) as well as graphical lasso (Figure 2). Both are valid methods, but it is unclear / confusing why the underlying metric for "distance between measures" is switched (i.e., also the correlations shown in Figure 1 / S2 could easily be visualized in a network plot; e.g. see references mentioned above for equivalent illustrations). Of course, both methods are absolutely valid and informative but it would be more consistent to stick to one method, or report a clear rationale for why such changes are made.

* Minor comment regarding terminology: I do not see how graphical lasso makes Figure 2 a "psychological graph" -- all figures in this paper are of course somehow "psychological graphs", and what F2 shows is simply one particular form of graph analysis.

* The MDS illustrations in Fig. 1 seem to imply two dimensions. How was this choice made? Moreover, how do these analyses (which are nowhere reported in detail) relate to the hierarchical clustering analyses?

* Ext. figure S2: How is it possible that R^2 's were smaller than 0? The authors' explanation in the methods section (stating that cross-validation might lead to this phenomenon) is not transparent enough.

* The authors rightly tested (and report) different regularizations for the predictive modeling analyses. They report having conducted analyses using non-linear methods, too. It is somewhat inconsistent that these results are not shown, and might make the impression of a file-drawer problem.

* Clustering analysis: I was not overly convinced by this part of the analysis, partly because it is not described sufficiently. The algorithm and the actual results should be described better (e.g., how do Figures 3 and 4 show exactly 13 clusters)?

Discussion:

* The authors conclude "For tasks in particular, EFA integrates multiple noisy DVs and creates stable measures of central psychological constructs." (l. 293). However, what does "stable" mean? One possibility is stability across tasks ("convergent validity"), another is stability across time ("reliability"). This is another example where the authors could build more strongly on past research and use existing concepts more precisely, in order to promote "cumulative science".

* On lines 277-279 the authors cite two papers regarding the ART, but these two papers do not investigate this task, at all.

References:

Costello, A. B., & Osborne, J. W. (2011). Best practices in exploratory factor analysis: Four recommendations for getting the most from your analysis. *Practical Assessment, Research and Evaluation*, 10(7), 1–9.

Eisenberg, I. W., Bissett, P. G., Canning, J. R., Dallery, J., Enkavi, A. Z., Whitfield-Gabrieli, S., ... Poldrack, R. A. (2017). Applying novel technologies and methods to inform the ontology of self-regulation. *Behaviour Research and Therapy*.
<https://doi.org/10.1016/j.brat.2017.09.014>

Frey, R., Pedroni, A., Mata, R., Rieskamp, J., & Hertwig, R. (2017). Risk preference shares the psychometric structure of major psychological traits. *Science Advances*, 3, e1701381.
<https://doi.org/10.1126/sciadv.1701381>

Harden, K. P., Kretsch, N., Mann, F. D., Herzhoff, K., Tackett, J. L., Steinberg, L., & Tucker-Drob, E. M. (2016). Beyond dual systems: A genetically-informed, latent factor model of behavioral and self-report measures related to adolescent risk-taking. *Developmental Cognitive Neuroscience*.
<https://doi.org/10.1016/j.dcn.2016.12.007>

Lilienfeld, S. O., Sauvigné, K. C., Lynn, S. J., Cautin, R. L., Litzman, R. D., & Waldman, I. D. (2015). Fifty psychological and psychiatric terms to avoid: a list of inaccurate, misleading, misused, ambiguous, and logically confused words and phrases. *Educational Psychology*, 1100.
<https://doi.org/10.3389/fpsyg.2015.01100>

Lönnqvist, J.-E., Verkasalo, M., Walkowitz, G., & Wichardt, P. C. (2015). Measuring individual risk

attitudes in the lab: Task or ask? An empirical comparison. *Journal of Economic Behavior & Organization*, 119, 254–266. <https://doi.org/10.1016/j.jebo.2015.08.003>

Stahl, C., Voss, A., Schmitz, F., Nuszbaum, M., Tüscher, O., Lieb, K., & Klauer, K. C. (2014). Behavioral components of impulsivity. *Journal of Experimental Psychology: General*, 143(2), 850–886. <https://doi.org/10.1037/a0033981>

White, J. L., Moffitt, T. E., Caspi, A., Bartusch, D. J., Needles, D. J., & Stouthamer-Loeber, M. (1994). Measuring impulsivity and examining its relationship to delinquency. *Journal of Abnormal Psychology*, 103(2), 192.

Reviewer #2 (Remarks to the Author):

The authors of this paper advocates for a data-driven approach to develop psychological ontologies – formal descriptions of concepts and their relationships in a given domain. They apply their approach to the multidisciplinary study of self-regulation – the goal-directed monitoring and modulation of thoughts, feelings, and behavior.

As a social psychologist who studies self-regulation, I will largely refrain from evaluating the details of the authors' "big data" analytical approach as much of it extends beyond my technical expertise. Nevertheless, let me observe that this study is incredible: the size of the participant sample, the sheer number and diversity of measures included, and the sophistication of the data analysis techniques are all "unprecedented" – to use the authors' own language. To be direct, I was blown away by the ambitiousness of the research goals and by the rigorous and advanced data-aggregation and analysis approach. I know of no other research study like this.

I will first briefly summarize what I see as the strengths and potential impact of this paper. Some of the most impactful and exciting conclusions one can draw from this research are its contributions to cognitive science. Perhaps the biggest "bombshell" of this work is the observation that cognitive performance variables long presumed to play a central role in behavior actually fail to predict any behavior even though they were rigorously assessed using state-of-the-art methods. This finding reminds me of Mischel's (1968) classic critique of the failure of personality traits to predict behavior, as well as the turmoil in attitudes research when it was suggested that attitudes do not predict behavior (e.g., Piere, 1934). Of course, traits and attitudes DO predict behavior – but conclusive evidence was provided only after refinements in measurement and theories to address the questions of when and how. Attitudes, for example, predict behavior when their assessment meets the compatibility principle (e.g., Ajzen & Fishbein, 1977), and when these attitudes are strong rather than weak (e.g., Krosnick & Petty, 1995). One could imagine the present paper serving as a similar wake-up call to cognitive scientists, leading them to question their long-held assumptions and spur the development of new measurement approaches and theoretical models.

Another important contribution to cognitive science is the dominance of the DDM (and discounting) parameters over latent constructs like "inhibition." That is, although inhibition is treated as THE central cognitive construct in self-regulation by psychologists (including those in social, cognitive, health, developmental, clinical and other subdisciplines), there is no evidence of this as a central parameter in the present work. Again, I imagine these data will serve as a wake-up call for many who have based their theoretical models on this construct. That discounting is not related to the survey measures and only weakly predict behavior may also spur similar theory re-visitation among those in the behavioral economics and judgment and decision-making tradition.

One disappointment I had with this paper was that there was very little discussion of the weaknesses of this data-driven approach. One weakness is that the method assumes constructs interact in a bivariate manner. To use the language of ANOVA, the authors' methods assume "main effects" and have trouble describing or accounting for more complex relationships (i.e., "interactions"). The weaknesses of this approach are evident in the analysis of the survey data. What should we make of the 12 factors that their analysis extracts? What exactly do we learn from pulling out "emotion control" from "eating control" in a manner that is independent of "reward sensitivity" and "goal-directedness." The authors suggest that their model examines the relationships between variables (that is what an "ontology" does), but does so by characterizing them in a specific manner – i.e., bivariate correlations. It struggles to account for relationships that may be more complex. The problem here is that there is good reason to expect more complex relationships. Eating control might be better related to goal-directedness and reward sensitivity if we knew whether or not people care about losing/maintaining weight and have a propensity to food rewards specifically. Eating control is entirely irrelevant to these two constructs – but it may be connected in a much more highly specific way than a simple bivariate correlation. Thus, a factor analysis might pull out eating control as being separate from these other two constructs, but that's because it assumes "main effects."

Note too that whether risk-taking should promote or impair self-regulation cannot be explained by simple bivariate relationships. Risk-taking is good when it comes to saving for retirement at a young age as one should invest in riskier stocks rather than more conservative bonds. But such risk-taking would be bad among those closer to retirement. Again, the authors data analytical approach is unlikely to be able to model such dynamics as it always assumes "main effects."

Thus, although I think this work has considerable strengths, it also has weaknesses which should be explicitly identified and discussed. My concern here is that this approach is particularly useful for evaluating "simple" relationships, but less so for more complex relationships. What I imagine will occur in response to these data, though, is that researchers will develop increasingly sophisticated models that describe these more complex relationships, thus ultimately diminishing this approach's utility. Some discussion of whether and how these analytical techniques might address some of these more complex associations thus deserves discussion. One could belittle the current work by suggesting that this data-driven approach is useful for ruling out "oversimplified" models, with the implication that people should be more sophisticated about their conceptual and theoretical models. That's fine, but one could have said that without data (although, I do agree that having data helps a lot!). Logically, the claim that some ability (as assessed by reaction time) would predict behavior is really silly. To observe self-regulation, you need a propensity for "bad" responses, some motivation to prevent those responses, and the ability to implement that latter motivation – i.e., at minimum, an interaction effect between 3 variables. Although having data to rule out a simple "main effect" is a useful argument, I am less convinced that this is absolutely necessary. Thus, this data driven approach does provide an empirical approach to the question of theory evaluation and integration – but only of a certain kind of theory.

Beyond complex interactions, the authors might also consider to what extent their data is impacted by the compatibility principle (Ajzen & Fishbein, 1977). I think that this is partly why domain specific survey measures do not map on that well to domain-general measures, as well as why performance task parameters do not predict behavior. If we can attribute weak or non-existent correlations to poor measurement, then does this data-driven approach really reveal the ontology that promotes cumulative science in the way that the authors claim it does? Or does it simply lead to re-evaluation of the measures and/or data-driven approach?

Reviewer #3 (Remarks to the Author):

Summary: Eisenberg et al. aim to do two things in this paper: using a large battery of self-control measures (including both tasks and surveys), they attempt to uncover the dimensional structure of the measured variables, and to assess how that dimensional structure relates to real-world self-reported outcomes of theoretical interest (e.g., smoking, obesity, etc.). To do this, the authors used an impressively thorough approach, combining dimension reduction and machine learning techniques to roughly 200 different dependent variables from 60 measures of self-regulation. Their results provided compelling evidence that surveys and tasks correlate weakly if at all and are best represented by two distinct psychological spaces, with 12 and 5 factors respectively. They also show that surveys predicted self-reported real-world outcomes (e.g., drug/alcohol use, physical/mental health) better than tasks, which had nearly zero predictive ability. Together, these findings lead the authors to conclude that self-regulation research might require a revised cognitive ontology, and that adopting data-driven approaches like theirs could clarify and improve research programs and lead to better behavioral interventions.

Overall evaluation: This is a methodologically impressive study, and I think it should serve as model for how to conduct open, integrative, reproducible, rigorous and methodologically innovative science. I also think that the goal of trying to link what are often far too disparate approaches and conceptualizations of self-control is an important one that the field needs to move forward. The findings are interesting, and I suspect they will generate useful debate about whether the somewhat pessimistic conclusions, particularly with regard to prediction of outcomes by tasks, are warranted. I think this study deserves publication. That said, I have more mixed feelings about the theoretical advancements made by the paper, which I elaborate on below. I thus have a few suggestions, as well as some questions, that I think the authors could consider in revising their manuscript.

1. The current data do a beautiful job showing that there is a mixed-to-null direct relationship between executive/inhibitory control task measures on the one hand and self-control surveys or real-world behavior on the other, and I think this paper will rightly be widely cited for demonstrating this so clearly. However, from the social psychological perspective, this finding is perhaps not wholly surprising. First, others have also recently reported similarly low correlations between executive control tasks and self-control surveys (Saunders, Milyavskya, Etz, Randles, & Inzlicht, in press, Collabra). Perhaps more importantly, psychologists have long theorized that there may not be a direct relationship between executive control performance and self-control/goal success, but rather that there may be an important interaction with contextual factors like implicit/automatic preferences, etc. (e.g. Hofmann, Friese & Strack, 2009, *Perspectives in Psychological Science*; Appetite; Nederkoorn et al., 2010, *Health Psychology*; Peeters et al., 2012, *Addiction*), stress (e.g., Quinn & Joormann, 2015, *Emotion*; Quinn & Joorman, *Clin Psych Sci*), or the amount of time on task (e.g., Nederkoorn et al., 2006). Similarly, some research suggests that inhibitory control needs to be tested in the context of hedonically activating stimuli to show relationships to real-world outcomes/behavior (e.g. Casey et al., 2011, *PNAS*; Houben et al., *Obesity*, 2013). Admittedly, some of this research suffers from the criticism raised by the current paper, in that it does not use the best predictive methods out there, does not use pre-registration, etc. So I look on this literature with some suspicion. Nevertheless, it raises the possibility that task-based control measures might relate to outcomes better after taking into account interactions with other factors, such as reward-related responding.

If I understood the methods correctly, the authors have tried to predict outcomes based only on direct, linear relationships of simple factor scores. I don't know whether it would be feasible to explore models in which there are interactions among survey factor scores and task-based factors of things like response caution or speeded IP, but it feels like, before concluding that factors measured by executive control tasks have no predictive power, one might want to think carefully about some of the

theories out there in the self-control literature and try to actually instantiate versions of them with the current data. This might in turn help to improve the theoretical advancement represented by the paper.

2. In a related vein, after having read the paper, I feel that I haven't learned quite as much as I had hoped, at a theoretical level, given the ambitions of the authors. For example:

a. The authors find that self-report measures and task measures don't correlate well, and self-report measures predict self-report measures of real-world outcomes better. Essentially, what the current study has shown is that survey measures predict survey measures better than cognitive tasks, which isn't wholly surprising given the psychometrics of surveys and cognitive tasks (i.e., robust behavioral tasks don't produce reliable individual differences; e.g., ref 37 cited by the authors), and the fact that survey responses may themselves be informed by a person's knowledge of their own behavior and outcomes (e.g., someone might think, "I drink heavily, therefore I am more like the definition of an impulsive person, therefore I should answer questions asking about my impulsivity in the affirmative.").

b. We learn that a set of questionnaires designed to measure constructs like impulsivity, sensation seeking, mindfulness, various kinds of risk taking, emotional control, etc. break down into facets that represent impulsivity, sensation seeking, mindfulness, various kinds of risk taking, and emotional control, etc. The novelty of the methods in finding this is interesting and important, and I think the ability to more reliably measure these factors is methodologically extremely important, but I'm not wholly sure what theoretical advance it represents. I am also left a bit unsure as to how much this factor structure is influenced by the specific set of measures the authors chose to include. Are there any factors present in the resulting reduction that are surprising given the inputs? Are there any absences of factors that are surprising (i.e., things we thought might be a "real" thing, but probably aren't)? Or is this simply the expected breakdown of factors, given the set of questionnaires that the authors included?

c. I think the authors try to address the previous question somewhat in their section on hierarchical clustering, but they could make clearer what added value the hierarchical clustering gives, especially for someone not fully versed in these methods. They state that they are trying to develop an understanding of "psychological constructs" but then the tree largely seems to recapitulate the factor structure (at least for surveys). Although the authors note some exceptions, and the task tree seems a bit more complex, I had some trouble figuring out what real-world or theoretical takeaways I can glean from this exercise. For example, what does it mean that there is a task cluster that loads on both strategic and speeded information processing? Why is this important? What current or future theories of behavior might it inform? I could have used more help from the authors here.

d. I found myself a bit disappointed in the survey prediction results and their discussion. Most outcomes still have fairly low cross-validated predictive accuracy, and the ones with the best predictive accuracy (obesity, mental health) seem almost circular. Specifically, surveys assessing eating control are the best predictors of outcomes related to eating behaviors/obesity. Surveys assessing poor emotional control are the best predictors of mental health outcomes, which when one looks closely turn out to be themselves essentially just versions of self-reported negative emotion (i.e., feeling depressed, hopeless, nervous, etc.). This feels almost like concluding that the definition of a problem predicts the problem, which isn't particularly theoretically novel. Could the authors speak to this issue?

e. Inspired by this observation, I also found myself wondering how the inclusion of specific surveys affects the conclusions. For instance, for obesity, where one has questionnaires relating to food measures, the strongest predictors are eating control surveys. But for drinking outcomes, where no survey or factor specifically relates to these kinds of issues, the behavior is predicted more by a range of other factors (reward sensitivity, ethical risk taking, etc.). I thus found myself wondering whether something like obesity would be predicted well by other factors if the factor specifying eating control was dropped from the predictive analysis, and whether we might learn a bit more about the

psychological origins of this outcome in the process.

f. In their discussion, the authors articulate a few things that we do discover (e.g., what the angling risk task actually assesses, what decision components contribute to SSRT) but these feel like relatively small advances for such a comprehensive dataset. I wanted to see a bit more of the bigger theoretical picture, especially as relates to this larger idea of "self-control."

3. The authors argue that using this cognitive ontology will give us a better chance of developing behavioral interventions, but I found myself wondering whether this is really true, given that the predictive accuracy of most models were fairly low, and the models with the best predictive accuracy may essentially be circular (see comment 2d above).

4. Finally, I found myself wondering whether the results from the authors actually represent a lower bound on the predictive accuracy of survey or task measures. Even though the authors have taken steps to ensure good data quality, their participants were left to complete about 10 hours of tasks/surveys at their convenience (with the only stipulation to complete within a week) via MTurk, where there may be many distractions, etc. This lack of control over participants' attentional or psychological states could affect data quality and/or predictive accuracy in many ways. I think the use of the more reliable factor scores might get around this concern at one level, but I wondered whether there might be issues at a higher level that would lead to essentially false negatives. Can the authors comment on this? Is there some room for optimism given this possibility?

5. Given that other research has sometimes reported links between task-based measures of e.g. inhibitory control and other self-control outcomes (e.g., Tabibnia et al., 2011, *J Neuro*; Lopez et al., 2014, *Psych Sci*; Berkman et al., 2011; *Psych Sci*), can the authors comment on how their results inform our interpretation of these studies? Do they see this previous work as just simply an example of the same issues psychology is facing in other domains (e.g., lack of preregistration, lack of open science, low replicability), or is there something else going on? What do we learn from their work about how to think about prior literature?

Some smaller points:

1. The results suggest that the various constructs in self-regulation research needs to be integrated much better. Self-regulation researchers are already aware of the conceptual muddle and have been trying to integrate different constructs in the last decade (e.g., Kelley, Wagner, & Heatherton, 2015, *Annu Rev Neurosci*; Kotabe & Hofmann, 2015, *Pers Psych Sci*; Hofmann, Friese, & Strack, 2009, *Pers Psych Sci*; Hofmann, Schmeichel, & Baddeley, 2012, *Trends Cog Sci*). Can the authors relate their findings to some of these theories, to make clearer the importance of their contributions?

2. Given how long it takes to collect these measures (~10 hours), do the authors have practical recommendations for how to use these techniques for researchers who don't have that much time or resources?

3. The authors should clarify and justify how and why these 60 surveys/tasks were chosen, especially the ones that aren't usually considered as standard measures of self-regulation. For example, it's not immediately clear why surveys like the Standard Leisure-Time Activity Categorical Item and Ten-Item Personality Inventory were included. Similarly, why was sentiment analysis of the Writing Task included? Are there previous studies showing relationships between self-regulation and these tasks? Especially given that the factor structure that is derived may be quite sensitive to what measures are included it would be nice to have some sense of how the authors determined the final battery of tests.

4. Did the authors have any measures of post-error slowing effects, which have been observed to correlate with self-regulation (Robinson, 2007, J Exp Social Psych)?
5. Line 46: Not everyone, especially social psychologists, will know exactly what you mean by "modern assessments of predictive accuracy." I take it you mean here things like machine learning with proper cross validation, etc., but you might want to spell this out.
6. The authors state in the text that there are 5 and 12 factors associated with surveys and tasks respectively (line 129 of main text), but I think maybe they have reversed the order? Shouldn't it be 12 and 5 for surveys and tasks?
7. Are the total number of DVs 196 or 205? The numbers are different in the main text (n = 196, line 78) and supplement (n = 205, line 74).
8. Given the potential issues with running experiments on Amazon MTurk, I think the authors should mention explicitly in the main text (line 74) that the data were collected from Amazon MTurk.
9. There are several typos in the supplement – the authors may want to go through them with a fine tooth comb before publication

Reviewers' comments:

Reviewer #1 (Remarks to the Author):

This paper reports a large study of 522 participants (recruited online on Amazon mTurk) who completed 23 questionnaires and 37 tasks related to "self-regulation". The authors make a general claim regarding cumulative psychological science, arguing that data-driven ontologies lay the groundwork to uncover mental structures. In particular, the authors argue that past psychological research a) suffered from siloed scientific traditions and b) focused on explanation rather than prediction, and claim that their approach -- in which self-regulation is almost used as a "case study" to illustrate this general point -- can overcome these issues. Their "data-driven ontology discovery" approach identified a strong gap between tasks and surveys. Further, the authors conclude that within the two classes of measures the ontology reveals "opportunities for theoretic synthesis" and that there are stable individual traits. Finally, surveys turned out to generalize moderately to (self-reported) real-life outcomes, whereas tasks did not.

This really is an exciting project, and I am particularly impressed by the effort that the authors have invested in setting up the entire online framework to assemble this large dataset, as well as by the numerous detailed and (mostly) informative analyses. Moreover, in line with current open science standards, the data-collection plan was pre-registered (but not the exact analysis plan), and the materials, data, and analysis code are openly available such that everything can be reproduced. Therefore, this study definitely deserves publication. That said, I have several conceptual questions and comments regarding the framing and scope of the current manuscript.

Specifically, the authors make a strong claim regarding cumulative science and the need for data-driven "ontology creation". In principle I agree with these points, but such an approach also has its risks. In an ideal world, applying a purely data-driven approach to a sufficiently large number of diverse measures might indeed reveal a series of clear-cut cognitive dimensions. Of course, to be truly informative such dimensions would have to be further validated (e.g., with imaging methods, by demonstrating substantial incremental predictive validity for specific life outcomes, etc). However, if such a validation fails or provides only moderate evidence, theory development is naturally very limited. So does the present study succeed in "ontology creation"?

For several reasons mentioned below, the "ontology" of self-regulation remains quite opaque to me after having read this article, despite this impressive and large data-set, and despite the richness of all the analyses. Rather, the reader is left with several open questions (regarding the ontology of self-regulation itself but also regarding the usefulness of a purely data-driven approach). Given that this approach, in my view, did not quite live up to the expectations that the

authors raised in the introduction, it is all the more a pity that there is no proper review of what is already known regarding (the ontology of) self-regulation and related constructs, which would have permitted contextualizing the present findings therein. This lack of integration into the current literature limits the impact of the present findings (which are informative and interesting even if the entire ontology remains somewhat unclear), which is somewhat paradoxical given the authors' strong claim for "cumulative science".

My general recommendation is therefore that the authors revise the article with a somewhat more "balanced" perspective, adopting a more critical view regarding how much we have actually learned about the ontology of self-regulation in this study, including a discussion of the risks of such a purely data-driven approach. Please find below a series of suggestions / and comments that I hope are useful to this end.

General comment:

1. The authors have already published an article in "Behavior Research and Therapy" (Eisenberg et al., 2017), in the context of which (preliminary) analyses have been conducted using the same sample (but no results are reported there), and where they claim to focus on "ontology creation" (which is, as I understood, the main goal of the present article). Even though this article is cited once in the present article, there is absolutely no discussion of how the two articles relate to each other. My reading is that the original article functions as kind of a pre-registration. If this is true, the authors should explain which changes in the analytic process they implemented, and why they did so. Moreover, the original article mentions additional components of this study that are not explained here, at all (imaging, mobile assessments, etc.).

That article is a protocol paper outlining the plan for a larger research program, of which this manuscript is one component. The OSF pre-registration is the proper pre-registration, as the 2017 paper does not include sufficient details to serve as such. We have taken steps to better clarify how our analysis deviated from that pre-registration (see *Deviations from pre-registration*).

Major conceptual points regarding the ontology / ontology discovery:

1. The authors hardly discuss the identified dimensions (i.e., the factors), even though they are supposed to be the core elements of an ontology (there are some further details in the supplemental materials and online, but the interpretation of these results is largely up to the reader). This is surprising and certainly does not help conveying what this ontology actually entails. Furthermore, it is somewhat unclear how strongly the inclusion of

particular measures influences which dimensions are eventually identified. Consequently it remains somewhat vague how strongly the identified dimensions are indeed part of the "ontology of self-regulation", particularly given the absence of a strong rationale for the measurement selection (including the set of "broader measures").

Thanks for this important conceptual point. We see dimensions and clusters as playing complementary roles in the ontology, and are agnostic about which one is more "core" to psychological theory. We have changed the general framing of the cognitive ontology to reflect this (first paragraph of *Ontology Creation*)

That said, we do believe that the factor labels are interpretively useful, and agree with the reviewer that our discussion was overly short. We have thus added additional discussion of the individual factors. We also feel our extended discussion of the clusters (in response to another of the reviewer's points) helps to strengthen the theoretical contribution of this article, and flesh out the components of our ontology.

As to the reviewer's point about the robustness of the particular factor solution, we have included two new robustness analyses for the factor models (3rd paragraph of *Creating a psychological space*). The first bootstraps our sample to create loading confidence intervals, while the second drops out single measures (with all of their constituent DVs) and refits the model comparing the solutions. We have also included robustness analyses for the clustering solutions (2nd paragraph in *Clustering within a psychological space*). We hope these analyses will assuage the reviewer's concerns regarding robustness.

2. Of course, a complicating factor in this respect is that the identified dimensions do not span both measurement classes. However, by now it is well established that there are substantial gaps between tasks and surveys of various (highly related) psychological constructs, potentially due to purely measurement-related reasons (e.g., Harden et al., 2016; Frey et al., 2017, which has almost an identical analysis and plot). In principle, this strong empirical finding does therefore not rule out the possibility that there could still be a clear ontology of self-regulation. I did not find it overly plausible to assume that this ontology consequently entails two separate "psychological spaces".

We agree that, in principle, there are connections between the constructs measured by surveys and tasks. "Psychological spaces" are conveniences to place measures within the context of other measures - to quantify psychological distance. Because the psychological graph is essentially disconnected (with two subgraphs), it ensures that the factor analytic results would bifurcate - resulting in a number of task specific factors and a number of survey specific factors. This is evident in Frey et al. 2017 (F7 independently and solely captures the behavioral tasks and the

higher-order R weakly loads on the behavioral tasks). Distances also become meaningless as all we can say is that tasks and surveys "are not close". Given this, we opted to analyze each modality separately. We have added a line to this effect in the text (2nd paragraph of *Creating a psychological space*). We have also highlighted (and included relevant citations) that this result has some precedent (1st paragraph of *Creating a psychological space*).

3. The transformation of the psychological spaces into clusters is an interesting approach. However, essentially nothing is done with these clusters here (the respective results section is rather a methods section, and only half a paragraph actually reports a few results, namely, that 13 clusters emerged, and a very brief interpretation of one branch and its two subbranches). It is left entirely open whether the ontology of self-regulation ultimately consists of the identified dimensions (factors) or rather the hierarchical clusters.

We thank the reviewer for asking for more here as well, as additional reflection on the nature of the cluster solutions proved fruitful. We have expanded our discussion of the clusters and updated our figures to reflect these new labels (*Clusters within the psychological space* and Figure 4/5)

4. Finally, whereas there was an analysis on cumulative R^2 's for the prediction of life outcomes, no analyses are reported regarding whether specific dimensions systematically relate to particular life outcomes, but not others. Furthermore, given that the factors were also correlated with each other, it remains unclear how "important" (and consequently, how robust) they actually are, and whether it is justified to consider them elements of the ontology of self-regulation.

To the reviewer's first point about whether specific dimensions systematically relate to particular life outcomes, we tried to address this in two ways: (1) through polar plots in the main text (Figure 6, 7) showing how ontological factors relate to each target outcome, and (2) in the extended figures (formerly Figure S8, now Figure 8 in the main text) to show how the range of target outcomes is related to each ontological factor - the inverse of the former. For instance, there you can see that the "reward sensitivity" factor is weakly related to all outcomes, the "eating control" survey is only related to the "obesity" factor, and that "impulsivity" is related to drinking, smoking and drug use. We have added this addition to the main text where we reference this figure.

The second point is interesting, and reflects a tradeoff in interpretability/usefulness of the dimensions. Given that there is some collinearity between our factors it is true that the polar plots do not reflect the entirety of the variance explained, as they are visualizing beta values. While orthogonal rotation would address this issue, we are unconvinced that factors should be

uncorrelated. Doing so produces other problems - for instance, the factors themselves become more complicated and less interpretable (whereas the oblique rotation attempts to optimize very simple structure). In addition, orthogonality seems an unachievable goal for psychology as a whole - different factorizations that perform useful theoretical work may be correlated. Given that the interpretability of predictive models rests on factor interpretability, we opted to optimize that.

5. In light of these points, what have we really learned regarding the ontology of self-regulation from this data-driven approach? I think a fair conclusion would be that this study provided a series of important incremental findings regarding self-regulation (which factors might underly this construct; how strongly do these dimensions cumulatively relate to life outcomes?), and corroborated the observation that tasks and surveys of psychological dimensions do not converge well (e.g., Lönnqvist et al., 2015; Harden et al., 2016; Frey et al., 2017). However, I do not think that the current approach really lives up to all the expectations the authors raise in the introduction. The findings would clearly be more informative were they better embedded in the existing literature (see next comment).

We agree that the work should be better situated within the existing literature. We have greatly expanded our discussion to describe connection between our work and previous work and have better highlighted how our approach (factor analysis) and results (task and survey bifurcation) are in line with some work in the literature. We have also described some high-level takeaways (*Beyond Self-Regulation*) and examples of specific theoretical advances (2nd paragraph of *Benefits of an Ontological Perspective*).

Integration with literature:

1. The authors make a strong claim for cumulative science and criticize siloed research traditions. However, they themselves hardly review the theories and empirical findings on self-regulation: What is known regarding self-regulation and its overlap with related constructs? And how has this past knowledge been obtained? What psychometric / modeling approaches do exist in the literature? One can certainly criticize aspects of past research, but the authors could differentiate a bit better how their approach is really novel, and in what respects it actually implements elements that have already been used in the past.

We agree that additional connections to the self-regulation literature would improve the paper. We have added additional discussion and citations in the *Discussion* section and the second paragraph of the *Introduction*. The extent of these additions were limited by journal word limits.

2. For instance, theories in the fields of personality or intelligence have been refined iteratively for about a century, using various measures and similar data-driven methods as implemented here. Moreover, there were assessments of the convergent and discriminant validity with other (potentially related) constructs and their predictive validity for real-life outcomes (e.g., White et al., 1994). It would be useful to discuss this research here.

Again here, we are limited by word count, but have tried to insert a bit more discussion throughout the manuscript.

3. Relatedly, I see the main novelty of the present study in the large and diverse dataset of self-regulation measures -- as it has been done in recent attempts for related constructs. However, most of the methods implemented here are quite established in psychometrics, and the "data-driven approach" is in fact not as novel as the article implies. By largely neglecting this literature and using their own terminology, the authors raise the expectations that the current approach is fundamentally different from previous methods (see also previous comment). For example, they claim to do "data-driven ontology development" based on "statistical techniques that capitalize on similarity between variables", and that this strategy is "derived from a classical approach in psychology, factor analysis". Why not call a spade a spade and use the established terminology, rather than raising the impression that some novel and magic methods are going to be implemented here?

We wished to distinguish the overall goal - a quantitative framework - from the specifics of the methodology used (which we now better define in the first paragraph of *Ontology Creation* - for instance, clearly outlining the two-step approach of dimensionality reduction followed by clustering). As the reviewer highlights, that goal came at the expense of connecting to established terminology. The 2nd paragraph of *Ontology Creation* now explicitly connects our approach to factor analysis. We also now employ terms like convergent/discriminant validity more liberally to highlight this connection. Finally, we describe the theoretical similarity between our ontology and Meehl's concept of a nomological network in the first paragraph of *Benefits of an Ontological Perspective*.

4. The same is true for the term "ontology", which seems to be used almost exclusively by the authors of this research group. According to the authors, an ontology specifies latent psychological constructs (minor comment: this is a misnomer, as a "construct" is latent by definition; it should either read a "latent variable" or a "[psychological] construct"; Lilienfeld et al., 2015) and their relationship to specific measures. Please explain how "ontology" differs from concepts such as a "nomological network" or more generally, the "psychometric structure" of a construct -- both of which are established terms in the literature and refer to how / whether various (cognitive) processes involved in particular constructs relate to each other (e.g., Stahl et al., 2014). Somewhat ironically, the authors seem to fall prey to a "jingle-fallacy" (l. 258) themselves due to the idiosyncratic (and in my view unnecessary) use of such specific terms.

We have further specified our meaning of an ontology, which we believe differs in its quantitative perspective from previous perspectives like nomological networks (1st paragraph of *Ontology Creation*). In addition, the term "ontology" is broadly used in other biological sciences, whereas a "nomological network" is particular to psychology. That said, we agree that our ontology is one instantiation of a nomological network that contextualizes measures and constructs in terms of other measures. As such, we have highlighted this connection in the discussion (1st paragraph of *Benefits of an Ontological Perspective*)

5. Besides this idiosyncratic use, the term "ontology" seems completely underspecified: The authors use the term in all possible ways, such as: ontology discovery, ontology creation, ontology construction, ontology development, cognitive ontology, data-driven ontology, ontological framework, ontological factors, ontological similarity, ontology revision, ontological fingerprints, ontological interventions. As such, this term becomes pretty much meaningless. For example, does an ontology inherently exist and needs to be discovered, or does an ontology need to be constructed / developed? Does it refer to the psychological structure or is an ontology data-driven? Of course, the authors are free to use whichever terms they prefer, but this is certainly not facilitating theory integration / development (cf., cumulative science!).

We found the point about ontology discovery vs. construction quite interesting! It challenged us to take a stance on the reality of this view of psychological measurement. We have removed the word "constructed", preferring "discovery". While we are humble about the "truth" of this particular ontology, we do believe it is a step closer to reflecting the actual organization of mental processes. We have made other changes including removing the term "ontological factors" (replaced with "factors"), and changed all uses of the phrase "cognitive ontology" to "psychological ontology". To the final point, we believe that the framing in terms of ontologies actually helps integrate psychological science into the broader set of biological sciences.

Methodological comments:

1. The present study clearly shines in terms of the number of implemented measures (60) and data-points per participant (196). However, in comparison the sample size is relatively small ($N = 522$), resulting in a subject-to-item ratio of 2.7:1 -- which seems very low for a data-driven exploratory approach. One (yet potentially outdated) rule-of-thumb has been to use subject-to-item ratios of no smaller than 40:1 (thus substantially larger), and simulation analyses have shown that already ratios of 20:1 lead to error rates "well above the field standard of .05" (Costello and Osborne, 2011). Is the extracted factor structure (i.e., "ontology") therefore indeed robust enough?

We have performed robustness assessments using a bootstrap approach and find that the loadings are indeed robust, as discussed above. We have referenced these analyses in the main text (3rd paragraph of *Creating a psychological space*) and included them in the supplemental information.

2. There was a substantial variability in the time intervals for the retest-assessment. How was this taken into account in the analyses? Moreover, how were the bootstrapped ICCs exactly constructed (the reference is an unpublished ms. but the details should be provided here, too).

While we agree the details of the test-retest results are important, a sufficiently thorough treatment of them are beyond the scope of this manuscript. Another paper (now a preprint - Enkavi et al, 2018) outlines this work in great detail, which we have referenced. That paper shows that reliability was not related to the retest delay. We have included a section called *Factor Score Robustness* that details our bootstrapped procedure.

3. The section on predictive validity starts with a criticism of what went wrong in past psychological research. While I agree with several points, I disagree that the methodology of the current study can overcome all of these issues and "allows for a generic evaluation of the state of behavioral prediction" (l. 209): On the one hand, there are of course also many researchers degrees of freedom here (e.g., selection of task and survey measures, selection of real-life measures, choices of how latent variables were extracted, etc.).

Though we had to select a finite number of measures, this selection process was independent of our data or eventual conclusions, and thus differs substantially from problematic uses of researcher degrees of freedom (e.g., p-hacking). Also, as mentioned in the text, we tried several

prediction models, but we report the results of each. We agree with the reviewer that "generic evaluation" was overselling our analyses and have removed the sentence.

For example, I could not find any information regarding the EFAs conducted for the real-life measures. Why were there exactly 9 factors? How independent were the different factors (in particular the different drinking and the different smoking factors might be highly correlated with each other). Factor inter-correlations (for tasks, surveys, and outcomes measures) should definitely be reported in the manuscript and not merely in the "online notebook"

(with explicit numbers instead of the colored correlation tables).

We have now included the factor correlations as a supplemental figure (annotated along with the heatmap colors; Extended Data Fig. 5). The smoking factors and drinking factors are correlated with each other between .5 and .6. The real-life measure factors were computed (and the number of factors chosen) using the same criteria as all other factor analyses (now clarified in the Results on pg 40). We have also included descriptions of the specific measures/questions we used for the outcomes in the supplement. To the point about measurement invariance between surveys and self-report, we have included discussion of this point in the "*Connecting psychological measurements to the real-world*" discussion section.

Furthermore, there is some obvious measurement invariance between the survey measures and the real-life measures, which are a) self-reported, too, and b) collected in the same assessment period (i.e., this is not really "prediction"). Therefore, the authors should be a bit more cautious here and not oversell this analysis.

As to the use of the word "prediction", we reserve it for cross-validation, which relates to "out of sample" prediction. Though this isn't prediction in the future, this use of the term is widespread in the statistics/machine learning field. We have also added a number of limitations to our discussion ("Limitations of our approach", final paragraphs of "*Connecting psychological measurement to real-world behavior*"), which help to address the reviewer's worry of overselling the analyses.

Analyses / results:

1. As a general comment, I found the various analyses very interesting and largely informative. However, as the exact analysis plan was not pre-registered and as there is no clear overview, some analyses in this paper (and the extended data section) feel quite exploratory. This could be improved with an overview of the analysis plan. Moreover, several analyses (in particular in the online materials) felt a bit like a "menu to choose

from" ("we did it like this but also like that"), leaving the reader puzzled why each of these analyses is relevant.

We have now described in the manuscript the ways in which the final analysis plan did not conform to the OSF pre-registration, and the reasons for these deviations (*Deviations from pre-registration*). We would like to highlight for the reviewer that while the analyses were indeed exploratory in the sense that they were not testing specific hypotheses, they were not exploratory in the more pejorative sense of "trying lots of things to find something that works", as the plan for exploration was driven by the overall goals outlined in the pre-registration. That is, we did not try a multitude of analytic approaches to investigate the data, but rather created an analytical plan to best describe and unpack the data.

To the point about several analyses feeling like a "menu to choose from", in many cases there was no obvious "right" approach for a particular analysis. Rather than simply picking a single method and running with it in these cases, our overall strategy has been to test whether the conclusions hold across multiple convergent approaches. In other cases (e.g. testing of prediction using raw measures rather than EFA factors), we report multiple analyses in order to show which particular approaches are the most optimal.

2. The following is an illustration of the previous comment: To visualize the "psychological space" the authors use both [partial] correlations between DVs (i.e., in the EVs, Figures 1 and S2) as well as graphical lasso (Figure 2). Both are valid methods, but it is unclear / confusing why the underlying metric for "distance between measures" is switched (i.e., also the correlations shown in Figure 1 / S2 could easily be visualized in a network plot; e.g. see references mentioned above for equivalent illustrations). Of course, both methods are absolutely valid and informative but it would be more consistent to stick to one method, or report a clear rationale for why such changes are made.

In Figure 1 we show the absolute value of the correlation between any two variables as that is what entered into the cluster analysis. In Figure S2 we show the raw correlations to demonstrate that the two measurement modalities are obviously disconnected - no analysis is actually needed beyond computing the correlation and visualizing the data. We felt that such a depiction may be more approachable for some readers who are not familiar with graphical lasso. Finally, we use graphical lasso to estimate the psychological graph because we believe that is the best method to create such a graph (Epskamp et al. 2018; Behav Res Methods). Essentially, Extended Fig. 2 was meant to make the point that surveys and tasks are unrelated, regardless of how you analyze the data.

3. Minor comment regarding terminology: I do not see how graphical lasso makes Figure 2 a "psychological graph" -- all figures in this paper are of course somehow "psychological graphs", and what F2 shows is simply one particular form of graph analysis.

Thank you for this suggestion. We have removed the phrase "psychological graph".

4. The MDS illustrations in Fig. 1 seem to imply two dimensions. How was this choice made? Moreover, how do these analyses (which are nowhere reported in detail) relate to the hierarchical clustering analyses?

MDS is used to visualize higher dimensional relationships, customarily in two dimensions. It tries to find the best 2D projection that preserves distances between DVs. As it is just a visualization technique of the correlation matrix, it is not itself used for any analyses. We have included a section in the Methods ("*Visualization using multidimensional-scaling*") that describes this process.

5. Ext. figure S2: How is it possible that R^2 's were smaller than 0? The authors' explanation in the methods section (stating that cross-validation might lead to this phenomenon) is not transparent enough.

The coefficient of determination ("r-squared") is not computed by squaring the correlation coefficient, but rather as $1 - (\text{residual SS})/(\text{total SS})$. When computed on the same data used to fit the model, then this is mathematically equivalent to the squared correlation coefficient, and cannot be negative. However, the equivalence between the coefficient of determination and the squared correlation coefficient requires that the sum of the residuals is zero, which will generally not be the case when one is predicting out of sample using crossvalidation. In cases where the fitted model performs worse than the mean of the out of sample data, then the residual sum of squares will exceed the total sum of squares and the coefficient of determination will be negative.

6. The authors rightly tested (and report) different regularizations for the predictive modeling analyses. They report having conducted analyses using non-linear methods, too. It is somewhat inconsistent that these results are not shown, and might make the impression of a file-drawer problem.

The nonlinear prediction results are reported in extended data table 1 (prediction using factor scores).

7. Clustering analysis: I was not overly convinced by this part of the analysis, partly because it is not described sufficiently. The algorithm and the actual results should be described better (e.g., how do Figures 3 and 4 show exactly 13 clusters)?

The clustering results have been described in greater detail and Fig 3 and 4 (now Fig 4 and 5) have been further described.

Discussion:

1. The authors conclude "For tasks in particular, EFA integrates multiple noisy DVs and creates stable measures of central psychological constructs." (l. 293). However, what does "stable" mean? One possibility is stability across tasks ("convergent validity"), another is stability across time ("reliability"). This is another example where the authors could build more strongly on past research and use existing concepts more precisely, in order to promote "cumulative science".

We have removed use of the ambiguous word "stable" in favor of "reliable".

2. On lines 277-279 the authors cite two papers regarding the ART, but these two papers do not investigate this task, at all.

Those articles referenced the BART (balloon analogue risk task), a close relative of the angling risk task. We have clarified that previous work does not explicitly test the ART, but other "related sequential risk-taking tasks".

References:

Costello, A. B., & Osborne, J. W. (2011). Best practices in exploratory factor analysis: Four recommendations for getting the most from your analysis. *Practical Assessment, Research and Evaluation*, 10(7), 1–9.

Eisenberg, I. W., Bissett, P. G., Canning, J. R., Dallery, J., Enkavi, A. Z., Whitfield-Gabrieli, S., ... Poldrack, R. A. (2017). Applying novel technologies and methods to inform the ontology of self-regulation. *Behaviour Research and Therapy*.
<https://doi.org/10.1016/j.brat.2017.09.014>

Frey, R., Pedroni, A., Mata, R., Rieskamp, J., & Hertwig, R. (2017). Risk preference shares the psychometric structure of major psychological traits. *Science Advances*, 3, e1701381.
<https://doi.org/10.1126/sciadv.1701381>

Harden, K. P., Kretsch, N., Mann, F. D., Herzhoff, K., Tackett, J. L., Steinberg, L., & Tucker-Drob, E. M. (2016). Beyond dual systems: A genetically-informed, latent factor model of behavioral and self-report measures related to adolescent risk-taking. *Developmental Cognitive Neuroscience*. <https://doi.org/10.1016/j.dcn.2016.12.007>

Lilienfeld, S. O., Sauvigné, K. C., Lynn, S. J., Cautin, R. L., Latzman, R. D., & Waldman, I. D. (2015). Fifty psychological and psychiatric terms to avoid: a list of inaccurate, misleading, misused, ambiguous, and logically confused words and phrases. *Educational Psychology*, 1100. <https://doi.org/10.3389/fpsyg.2015.01100>

Lönnqvist, J.-E., Verkasalo, M., Walkowitz, G., & Wichardt, P. C. (2015). Measuring individual risk attitudes in the lab: Task or ask? An empirical comparison. *Journal of Economic Behavior & Organization*, 119, 254–266. <https://doi.org/10.1016/j.jebo.2015.08.003>

Stahl, C., Voss, A., Schmitz, F., Nuszbaum, M., Tüscher, O., Lieb, K., & Klauer, K. C. (2014). Behavioral components of impulsivity. *Journal of Experimental Psychology: General*, 143(2), 850–886. <https://doi.org/10.1037/a0033981>

White, J. L., Moffitt, T. E., Caspi, A., Bartusch, D. J., Needles, D. J., & Stouthamer-Loeber, M. (1994). Measuring impulsivity and examining its relationship to delinquency. *Journal of Abnormal Psychology*, 103(2), 192.

Reviewer #2 (Remarks to the Author):

The authors of this paper advocates for a data-driven approach to develop psychological ontologies – formal descriptions of concepts and their relationships in a given domain. They apply their approach to the multidisciplinary study of self-regulation – the goal-directed monitoring and modulation of thoughts, feelings, and behavior.

As a social psychologist who studies self-regulation, I will largely refrain from evaluating the details of the authors’ “big data” analytical approach as much of it extends beyond my technical expertise. Nevertheless, let me observe that this study is incredible: the size of the participant sample, the sheer number and diversity of measures included, and the sophistication of the data analysis techniques are all “unprecedented” – to use the authors’ own language. To be direct, I was blown away by the ambitiousness of the research goals and by the rigorous and advanced

data-aggregation and analysis approach. I know of no other research study like this.

I will first briefly summarize what I see as the strengths and potential impact of this paper. Some of the most impactful and exciting conclusions one can draw from this research are its contributions to cognitive science. Perhaps the biggest “bombshell” of this work is the observation that cognitive performance variables long presumed to play a central role in behavior actually fail to predict any behavior even though they were rigorously assessed using state-of-the-art methods. This finding reminds me of Mischel’s (1968) classic critique of the failure of personality traits to predict behavior, as well as the turmoil in attitudes research when it was suggested that attitudes do not predict behavior (e.g., Piore, 1934). Of course, traits and attitudes DO predict behavior – but conclusive evidence was provided only after refinements in measurement and theories to address the questions of when and how. Attitudes, for example, predict behavior when their assessment meets the compatibility principle (e.g., Ajzen & Fishbein, 1977), and when these attitudes are strong rather than weak (e.g., Krosnick & Petty, 1995). One could imagine the present paper serving as a similar wake-up call to cognitive scientists, leading them to question their long-held assumptions and spur the development of new measurement approaches and theoretical models.

Another important contribution to cognitive science is the dominance of the DDM (and discounting) parameters over latent constructs like “inhibition.” That is, although inhibition is treated as THE central cognitive construct in self-regulation by psychologists (including those in social, cognitive, health, developmental, clinical and other subdisciplines), there is no evidence of this as a central parameter in the present work. Again, I imagine these data will serve as a wake-up call for many who have based their theoretical models on this construct. That discounting is not related to the survey measures and only weakly predict behavior may also spur similar theory re-visitation among those in the behavioral economics and judgment and decision-making tradition.

One disappointment I had with this paper was that there was very little discussion of the weaknesses of this data-driven approach. One weakness is that the method assumes constructs interact in a bivariate manner. To use the language of ANOVA, the authors’ methods assume “main effects” and have trouble describing or accounting for more complex relationships (i.e., “interactions”). The weaknesses of this approach are evident in the analysis of the survey data. What should we make of the 12 factors that their analysis extracts? What exactly do we learn from pulling out “emotion control” from “eating control” in a manner that is independent of “reward sensitivity” and “goal-directedness.” The authors suggest that their model examines the relationships between variables (that is what an “ontology” does), but does so by characterizing them in a specific manner – i.e., bivariate correlations. It struggles to account for relationships that may be more complex. The problem here is that there is good reason to expect more

complex relationships. Eating control might be better related to goal-directedness and reward sensitivity if we knew whether or not people care about losing/maintaining weight and have a propensity to food rewards specifically. Eating control is entirely irrelevant to these two constructs – but it may be connected in a much more highly specific way than a simple bivariate correlation. Thus, a factor analysis might pull out eating control as being separate from these other two constructs, but that’s because it assumes “main effects.”

The simplicity of the relationships underlying our ontology (bivariate relationships) is indeed a limitation. Even within bivariate limitations more complex analyses could be performed (given more data). We agree that we did not adequately highlight these limitations and have included a section describing them (*Limitations of our Approach*)

As for the simple relationships underlying our predictive method - we tried random forests and have now also tried generalized additive models (though did not add this to the paper, as it did not meaningfully change any takeaways), both of which explore the possibility of non-linear relationships between measures and outcomes. None of these methods performed better than the simple linear ridge regression model.

Note too that whether risk-taking should promote or impair self-regulation cannot be explained by simple bivariate relationships. Risk-taking is good when it comes to saving for retirement at a young age as one should invest in riskier stocks rather than more conservative bonds. But such risk-taking would be bad among those closer to retirement. Again, the authors data analytical approach is unlikely to be able to model such dynamics as it always assumes “main effects.”

It's true that we do not have access to the social context that almost certainly modulates these relationships. Our findings should be taken as reflective of the "uncontextualized" relationship or "average relationship" across contexts. In the specific case mentioned, it's possible that age interacts with risk-taking to predict financial success (for example). These kinds of analyses seem better left to domain experts who may use this dataset, rather than addressing them in this paper. This is one of our purposes in making the data openly available. We have highlighted the need to evaluate environmental context, and possible next steps, in the final paragraph of the section *Connecting psychological measurement to real-world behavior*

Thus, although I think this work has considerable strengths, it also has weaknesses which should be explicitly identified and discussed. My concern here is that this approach is particularly useful for evaluating “simple” relationships, but less so for more complex relationships. What I imagine will occur in response to these data, though, is that researchers will develop increasingly sophisticated models that describe these more complex relationships, thus ultimately diminishing this approach’s utility. Some discussion of whether and how these analytical techniques might

address some of these more complex associations thus deserves discussion. One could belittle the current work by suggesting that this data-driven approach is useful for ruling out “oversimplified” models, with the implication that people should be more sophisticated about their conceptual and theoretical models. That’s fine, but one could have said that without data (although, I do agree that having data helps a lot!). Logically, the claim that some ability (as assessed by reaction time) would predict behavior is really silly. To observe self-regulation, you need a propensity for “bad” responses, some motivation to prevent those responses, and the ability to implement that latter motivation – i.e., at minimum, an interaction effect between 3 variables. Although having data to rule out a simple “main effect” is a useful argument, I am less convinced that this is absolutely necessary. Thus, this data driven approach does provide an empirical approach to the question of theory evaluation and integration – but only of a certain kind of theory.

We respectfully disagree with the reviewer that simple bivariate relationships are necessarily "oversimplified" models. Many constructs in the field are created or discussed precisely because they should have a simple relationship to real-world outcomes (e.g., stop-signal reaction time as an index of response inhibition, or temporal discounting rate, both of which have been claimed to directly relate to real-world behaviors.) The prevalence of these theories both points to (1) the potential reality that these models are not "oversimplified", but rather just "simple", and (2) the utility in synthesizing such frameworks regardless of their empirical validity, as they make up a large portion of the literature. We hope that large scale analysis of bivariate correlations will summarize these types of theories, and provide a structure that may inform further investigation.

That said, of course more complicated relationships involving the interactions of 3 or more variables are possible. Future directed investigation of particular phenomena affords the luxury of more sophisticated analyses. However, scaling such analyses up seems currently implausible (and may even be inadvisable given the sample sizes of many existing studies!). While datasets are relatively limited (even ours is not large enough for more sophisticated unsupervised learning techniques), we must trade off variance for analytical bias - here materialized as limiting the types of relationships we consider. We have discussed the limitations you bring up in the discussion at multiple points (see *Limitations of our approach* as well as the new paragraphs at the end of *connecting psychological measurement to real-world behavior*).

Beyond complex interactions, the authors might also consider to what extent their data is impacted by the compatibility principle (Ajzen & Fishbein, 1977). I think that this is partly why domain specific survey measures do not map on that well to domain-general measures, as well as why performance task parameters do not predict behavior. If we can attribute weak or non-existent correlations to poor measurement, then does this data-driven approach really reveal

the oncology that promotes cumulative science in the way that the authors claim it does? Or does it simply lead to re-evaluation of the measures and/or data-driven approach?

We believe it does both. To the extent that there is structure in the data, the approach reveals it, and this revealed structure is sensible. However, this structure is, as the reviewer says, limited by our data. Our statement is less about what the true ontology of psychological phenomenon should be, but rather what structure deserves our epistemic commitment. Given that the data are the way they are, we believe this work reflects what our ontology should be at this time. We have attempted to widen our scope of integration in comparison to previous approaches via a larger battery, but are still limited in the ways highlighted by the reviewer. We have emphasized some of these limitations in our expanded discussion (see *Limitations of our approach* as well as the new paragraphs at the end of *connecting psychological measurement to real-world behavior*).

Reviewer #3 (Remarks to the Author):

Summary: Eisenberg et al. aim to do two things in this paper: using a large battery of self-control measures (including both tasks and surveys), they attempt to uncover the dimensional structure of the measured variables, and to assess how that dimensional structure relates to real-world self-reported outcomes of theoretical interest (e.g., smoking, obesity, etc.). To do this, the authors used an impressively thorough approach, combining dimension reduction and machine learning techniques to roughly 200 different dependent variables from 60 measures of self-regulation. Their results provided compelling evidence that surveys and tasks correlate weakly if at all and are best represented by two distinct psychological spaces, with 12 and 5 factors respectively. They also show that surveys predicted self-reported real-world outcomes (e.g., drug/alcohol use, physical/mental health) better than tasks, which had nearly zero predictive ability. Together, these findings lead the authors to conclude that self-regulation research might require a revised cognitive ontology, and that adopting data-driven approaches like theirs could clarify and improve research programs and lead to better behavioral interventions.

Overall evaluation: This is a methodologically impressive study, and I think it should serve as model for how to conduct open, integrative, reproducible, rigorous and methodologically innovative science. I also think that the goal of trying to link what are often far too disparate approaches and conceptualizations of self-control is an important one that the field needs to move forward. The findings are interesting, and I suspect they will generate useful debate about whether the somewhat pessimistic conclusions, particularly with regard to prediction of outcomes by tasks, are warranted. I think this study deserves publication. That said, I have more mixed feelings about the theoretical advancements made by the paper, which I elaborate on below. I thus have a few suggestions, as well as some questions, that I think the authors could

consider in revising their manuscript.

1. The current data do a beautiful job showing that there is a mixed-to-null direct relationship between executive/inhibitory control task measures on the one hand and self-control surveys or real-world behavior on the other, and I think this paper will rightly be widely cited for demonstrating this so clearly. However, from the social psychological perspective, this finding is perhaps not wholly surprising. First, others have also recently reported similarly low correlations between executive control tasks and self-control surveys (Saunders, Milyavskya, Etz, Randles, & Inzlicht, in press, Collabra). Perhaps more importantly, psychologists have long theorized that there may not be a direct relationship between executive control performance and self-control/goal success, but rather that there may be an important interaction with contextual factors like implicit/automatic preferences, etc. (e.g. Hofmann, Friese & Strack, 2009, *Perspectives in Psychological Science*; Appetite; Nederkoorn et al., 2010, *Health Psychology*; Peeters et al., 2012, *Addiction*), stress (e.g., Quinn & Joormann, 2015, *Emotion*; Quinn & Joorman, *Clin Psych Sci*), or the amount of time on task (e.g., Nederkoorn et al., 2006). Similarly, some research suggests that inhibitory control needs to be tested in the context of hedonically activating stimuli to show relationships to real-world outcomes/behavior (e.g. Casey et al., 2011, *PNAS*; Houben et al., *Obesity*, 2013). Admittedly, some of this research suffers from the criticism raised by the current paper, in that it does not use the best predictive methods out there, does not use pre-registration, etc. So I look on this literature with some suspicion. Nevertheless, it raises the possibility that task-based control measures might relate to outcomes better after taking into account interactions with other factors, such as reward-related responding.

We agree that contextualizing factors are important and impacts how the prediction failure is interpreted. We have added discussion of these points in our expanded discussion. (see paragraphs at the end of *connecting psychological measurement to real-world behavior*).

If I understood the methods correctly, the authors have tried to predict outcomes based only on direct, linear relationships of simple factor scores. I don't know whether it would be feasible to explore models in which there are interactions among survey factor scores and task-based factors of things like response caution or speeded IP, but it feels like, before concluding that factors measured by executive control tasks have no predictive power, one might want to think carefully about some of the theories out there in the self-control literature and try to actually instantiate versions of them with the current data. This might in turn help to improve the theoretical advancement represented by the paper.

It is possible that prediction may be more successful with a more flexible model class. We tried random forests, and have now also tried generalized additive models (though did not add this to

the paper, as it did not meaningfully change any takeaways), both of which allow non-linear relationships between measures and outcomes. The reviewer mentions interactions between surveys and task factors as one addition we didn't explore, but due to the near-zero correlation between task and factor scores, such an interactive model would necessarily be no more effective than a non-interactive model including both task and survey factors (which we have already included). We also did look at random-forests which allow non-linear relationships between tasks and surveys, but this model performed poorly.

That said, perhaps more hypothesis driven tests of interactions between multiple factors (perhaps related to theories that posit some interaction between immediate and long-term reward and conflict resolution) would be successful. However, such models were beyond the scope of this paper, and scaling up such analyses to the full set of possible interactive models is currently implausible given our dataset. We hope that other researchers can make use of this dataset in a restricted, directed way, to speak to such models.

We have discussed limitations related to this discussion at multiple points (see *Limitations of our approach* as well as the new paragraphs at the end of *connecting psychological measurement to real-world behavior*).

2. In a related vein, after having read the paper, I feel that I haven't learned quite as much as I had hoped, at a theoretical level, given the ambitions of the authors. For example:

a. The authors find that self-report measures and task measures don't correlate well, and self-report measures predict self-report measures of real-world outcomes better. Essentially, what the current study has shown is that survey measures predict survey measures better than cognitive tasks, which isn't wholly surprising given the psychometrics of surveys and cognitive tasks (i.e., robust behavioral tasks don't produce reliable individual differences; e.g., ref 37 cited by the authors), and the fact that survey responses may themselves be informed by a person's knowledge of their own behavior and outcomes (e.g., someone might think, "I drink heavily, therefore I am more like the definition of an impulsive person, therefore I should answer questions asking about my impulsivity in the affirmative.").

We agree that this method relationship may partially explain the survey prediction results. If this is taken as a criticism, it is a criticism of survey design/interpretation generally. This strikes at the heart of survey validity - perhaps they are not measures of impulsivity but self-perception of impulsivity or self-concept, which is itself dependent on their behaviors, as suggested. We have highlighted this potential interpretation and used the reviewer's own example (much appreciated!). See paragraph 3 of *connecting psychological measurement to real-world behavior*.

However, while the surveys predicting moderately well (and better than tasks) is probably not surprising to most people, the almost complete failure of the tasks is noteworthy and challenges existing theory. We see this finding as important regardless of the relative success of surveys.

b. We learn that a set of questionnaires designed to measure constructs like impulsivity, sensation seeking, mindfulness, various kinds of risk taking, emotional control, etc. break down into facets that represent impulsivity, sensation seeking, mindfulness, various kinds of risk taking, and emotional control, etc. The novelty of the methods in finding this is interesting and important, and I think the ability to more reliably measure these factors is methodologically extremely important, but I'm not wholly sure what theoretical advance it represents. I am also left a bit unsure as to how much this factor structure is influenced by the specific set of measures the authors chose to include. Are there any factors present in the resulting reduction that are surprising given the inputs? Are there any absences of factors that are surprising (i.e., things we thought might be a "real" thing, but probably aren't)? Or is this simply the expected breakdown of factors, given the set of questionnaires that the authors included?

We agree that the survey factors *are* in line with expectations, and in that way the survey portion of this work is a conceptual replication of a large set of perspectives instantiated in various measures. A conceptual replication at this scale was not guaranteed (as is evident by the more surprising structure derived from the task EFA). That said, our factor structure is smaller than the number of measures (which presumably, to some researchers, each represent a discriminant factor that could have justified its own factor), and at least one factor (goal-directedness) integrates across multiple measures that, while sensible, were not a-priori obvious. We also agree completely that the factor structure necessarily is related to the specific set of tasks included in the battery, and now have included analyses demonstrating the factor structure's robustness to the removal of individual measures.

c. I think the authors try to address the previous question somewhat in their section on hierarchical clustering, but they could make clearer what added value the hierarchical clustering gives, especially for someone not fully versed in these methods. They state that they are trying to develop an understanding of "psychological constructs" but then the tree largely seems to recapitulate the factor structure (at least for surveys). Although the authors note some exceptions, and the task tree seems a bit more complex, I had some trouble figuring out what real-world or theoretical takeaways I can glean from this exercise. For example, what does it mean that there is a task cluster that loads on both strategic and speeded information processing? Why is this important? What current or future theories of behavior might it inform? I could have used more help from the authors here.

We have expanded this section (*Clustering within the psychological space*). What the reviewer picks up is true - the survey clusters and factors are almost identical. This is due to the nature of the survey factor structure - it has limited cross-loadings and exceedingly "very simple structure", which makes factors and clustering largely equivalent. That said, there is at least one branch of two clusters (self-control) that reflects a more complicated factor profile. Additionally, the task structure was much more complicated, indicating the utility of the method in identifying structure beyond the underlying factors.

d. I found myself a bit disappointed in the survey prediction results and their discussion. Most outcomes still have fairly low cross-validated predictive accuracy, and the ones with the best predictive accuracy (obesity, mental health) seem almost circular. Specifically, surveys assessing eating control are the best predictors of outcomes related to eating behaviors/obesity. Surveys assessing poor emotional control are the best predictors of mental health outcomes, which when one looks closely turn out to be themselves essentially just versions of self-reported negative emotion (i.e., feeling depressed, hopeless, nervous, etc.). This feels almost like concluding that the definition of a problem predicts the problem, which isn't particularly theoretically novel. Could the authors speak to this issue?

We have included discussion of exactly this circularity. We see this as a general issue with surveys, and not specific to this paper, but agree that it puts into better context the relative "success" of surveys compared to tasks. See paragraph 3 of *connecting psychological measurement to real-world behavior*.

e. Inspired by this observation, I also found myself wondering how the inclusion of specific surveys affects the conclusions. For instance, for obesity, where one has questionnaires relating to food measures, the strongest predictors are eating control surveys. But for drinking outcomes, where no survey or factor specifically relates to these kinds of issues, the behavior is predicted more by a range of other factors (reward sensitivity, ethical risk taking, etc.). I thus found myself wondering whether something like obesity would be predicted well by other factors if the factor specifying eating control was dropped from the predictive analysis, and whether we might learn a bit more about the psychological origins of this outcome in the process.

It's certainly possible that without "eating control" obesity would still be reasonably predicted by some combination of the other factors, whose contributions are all highly correlated with the eating-control survey. However, the large beta value for eating-control indicates that it contributes a substantial amount of unique variance. In general, we find it problematic to remove specific measures at this point to better articulate "psychological origins" that conform to a priori ideas of a "good" psychological explanation. Our analytic approach found that eating control was

separable from constructs defined by the other factors, and thus should be considered a part of a person's psychological phenotype.

f. In their discussion, the authors articulate a few things that we do discover (e.g., what the angling risk task actually assesses, what decision components contribute to SSRT) but these feel like relatively small advances for such a comprehensive dataset. I wanted to see a bit more of the bigger theoretical picture, especially as relates to this larger idea of “self-control.”

We meant our discussion to expose the utility of our framework at multiple levels. Theory advancement may take the form of specific claims about specific psychological phenomenon, and we wanted to highlight that potential. That said, we have greatly expanded our discussion - both in our summary of this "multi-level" point, and by including larger takeaways about self-regulation more generally (*Beyond Self-Regulation*).

3. The authors argue that using this cognitive ontology will give us a better chance of developing behavioral interventions, but I found myself wondering whether this is really true, given that the predictive accuracy of most models were fairly low, and the models with the best predictive accuracy may essentially be circular (see comment 2d above).

We share the reviewer’s skepticism that the approach will provide novel targets for intervention, but we feel that demonstrating a lack of relationship may also have significant value, by suggestion which potential intervention targets may be dead ends.

4. Finally, I found myself wondering whether the results from the authors actually represent a lower bound on the predictive accuracy of survey or task measures. Even though the authors have taken steps to ensure good data quality, their participants were left to complete about 10 hours of tasks/surveys at their convenience (with the only stipulation to complete within a week) via MTurk, where there may be many distractions, etc. This lack of control over participants' attentional or psychological states could affect data quality and/or predictive accuracy in many ways. I think the use of the more reliable factor scores might get around this concern at one level, but I wondered whether there might be issues at a higher level that would lead to essentially false negatives. Can the authors comment on this? Is there some room for optimism given this possibility?

One would hope that you are correct, but we have no reason to believe that the results of our outcome measures are substantially less reliable than others (see our other paper on PsArxiv: Enkavi et al, 2018). Our predictors (the factor scores) also show high reliability, as you mention, even if the constituent DVs sometimes do not. As such we believe our data should be taken as a representation of predictive power, at this time, given normal methods in psychology, rather than

a lower bound. We have no doubt that some strategies (especially ones not yet developed) may improve predictive accuracy.

5. Given that other research has sometimes reported links between task-based measures of e.g. inhibitory control and other self-control outcomes (e.g., Tabibnia et al., 2011, *J Neuro*; Lopez et al., 2014, *Psych Sci*; Berkman et al., 2011; *Psych Sci*), can the authors comment on how their results inform our interpretation of these studies? Do they see this previous work as just simply an example of the same issues psychology is facing in other domains (e.g., lack of preregistration, lack of open science, low replicability), or is there something else going on? What do we learn from their work about how to think about prior literature?

As the reviewer suggests, we believe this conflict is mostly likely reflective of the same issues psychology has been grappling with in other domains. We have included this explicitly at the end of the second paragraph "Connecting psychological measurement to real-world behavior"

Some smaller points:

1. The results suggest that the various constructs in self-regulation research needs to be integrated much better. Self-regulation researchers are already aware of the conceptual muddle and have been trying to integrate different constructs in the last decade (e.g., Kelley, Wagner, & Heatherton, 2015, *Annu Rev Neurosci*; Kotabe & Hofmann, 2015, *Pers Psych Sci*; Hofmann, Friese, & Strack, 2009, *Pers Psych Sci*; Hofmann, Schmeichel, & Baddeley, 2012, *Trends Cog Sci*). Can the authors relate their findings to some of these theories, to make clearer the importance of their contributions?

We have added a section in the discussion titled *Beyond Self-Regulation* that speaks to how our work relates to previous theories.

2. Given how long it takes to collect these measures (~10 hours), do the authors have practical recommendations for how to use these techniques for researchers who don't have that much time or resources?

We believe the general framework of dimensionality reduction followed by clustering is a helpful analytical strategy for many researchers, even if their measurement batteries are smaller than ours. Data-sharing is also, of course, another way for others to benefit. We hope that large datasets like ours will become more commonplace and widely available in the future.

Separately, we are working on articulating methods to allow other researchers to project their measures into a quantitative framework (ours or others), hopefully allowing other researchers to

quantitatively benefit from large studies like this without running similar studies themselves. This is work in progress, and is beyond the scope of the current manuscript.

3. The authors should clarify and justify how and why these 60 surveys/tasks were chosen, especially the ones that aren't usually considered as standard measures of self-regulation. For example, it's not immediately clear why surveys like the Standard Leisure-Time Activity Categorical Item and Ten-Item Personality Inventory were included. Similarly, why was sentiment analysis of the Writing Task included? Are there previous studies showing relationships between self-regulation and these tasks? Especially given that the factor structure that is derived may be quite sensitive to what measures are included it would be nice to have some sense of how the authors determined the final battery of tests.

We chose a set of measures that were generally used in the field and of broad significance. The specific examples mentioned by the reviewer *are* somewhat outliers. The Ten-item personality was included because we felt connecting with the large personality literature was worth 10 questions. The writing task was the most experimental, and was included (1) to allow sentiment analysis, which we thought may reflect the person's outlook around the time of completing the battery and (2) may hold additional promise moving forward when other NLP methods are applied. Removing the writing measures do not affect the factor structure at all.

The Stanford Leisure-Time Activity survey was meant to be an outcome measure (our best measure of physical activity), but was inadvertently combined with surveys. This has been corrected.

4. Did the authors have any measures of post-error slowing effects, which have been observed to correlate with self-regulation (Robinson, 2007, J Exp Social Psych)?

We did not calculate such a measure, though it is possible. We like this suggestion, but given that our measures were (largely) identified before the analytic framework was applied, we will leave this for subsequent work, along with other exploration of sequential effects.

5. Line 46: Not everyone, especially social psychologists, will know exactly what you mean by “modern assessments of predictive accuracy.” I take it you mean here things like machine learning with proper cross validation, etc., but you might want to spell this out.

Cross-validation is definitely the most important concept. We have included a parenthetical expression making this clear.

6. The authors state in the text that there are 5 and 12 factors associated with surveys and tasks respectively (line 129 of main text), but I think maybe they have reversed the order? Shouldn't it be 12 and 5 for surveys and tasks?

Indeed, thanks!

7. Are the total number of DVs 196 or 205? The numbers are different in the main text (n = 196, line 78) and supplement (n = 205, line 74).

The number of DVs was reduced from 205 to 196 after dropping out highly correlated variables and variables that could not be suitably transformed (as described in the section *Data Cleaning and imputation*). This has been better spelled out. Our final count has also gone down by three now (to 193) due to the LCAT correction based on your prior comment, and a correction to our preprocessing (we were removing outliers and assessing variable correlations before variable transformation, which we have changed to after variable transformation.)

8. Given the potential issues with running experiments on Amazon MTurk, I think the authors should mention explicitly in the main text (line 74) that the data were collected from Amazon MTurk.

Done. We also describe the possible effects of this choice in *Limitations of our approach*.

9. There are several typos in the supplement – the authors may want to go through them with a fine tooth comb before publication

Thank you for pointing this out. We have gone through the supplement and corrected a number of typos.

****REVIEWERS' COMMENTS:**

Reviewer #1 (Remarks to the Author):

The authors did a great job in addressing the comments, and it was a pleasure to read the much improved manuscript. I am convinced that this paper makes a valuable contribution to the field and it should therefore be published. I only have a few minor comments:

* The authors write: "This choice [of broad / diverse measures] affords the possibility of identifying the "borders" of self-regulation as a construct or rejecting the discriminant validity of self-regulation in itself." There is already some theorizing related to this in the discussion section, but this point could and should be picked up explicitly. It would be nice if the authors could give a specific answer / conclusion concerning this issue.

* Concerning the two separate psychological spaces the authors argue that "Future reconciliation of these two spaces may be possible." This should also be picked up in the discussion section, and it would be interesting to learn more about the authors' ideas of how one could potentially achieve this.

* Figure 3 (A / C): Instead of showing symmetric correlation matrices (with retest correlations only), it would be helpful if one of the two triangles showed factor-intercorrelations. Moreover, in addition to the colors actual correlation coefficients would be informative.

Reviewer #2 (Remarks to the Author):

I served as a reviewer during the previous round of reviews. I was very impressed with the paper and was generally supportive of it being published. My primary concerns rested on a lack of any discussion of the weaknesses of the authors' approach. The present revision now explicitly discusses these issues. Although I might quibble with a few of their minor arguments, I believe the authors have by-and-large adequately addressed my concerns.

Reviewer #3 (Remarks to the Author):

The authors have largely addressed my suggestions, and I am happy to recommend publication. The contributions of the paper are now much clearer. The Discussion section especially delivers more interesting ideas and fruitful reflection for the field.

I have only two very small remaining suggestions, neither of which are crucial:

1. In the Discussion, in the section in which the limitations of linear bivariate relationships is addressed, it might be useful to concretize this by an example for the reader, with reference to a related psychological literature. For example, I have in mind something like citing the idea that EF may only matter in cases of strong reward sensitivity, etc.

2. If there is space, I think the point in response to my previous comment 2E about the eating control surveys being separable from other factors and perhaps being best considered an independent part of a person's psychological phenotype is a good one, and could perhaps be made directly in the paper somewhere (either in the Results or Discussion section).

Reviewer #1 (Remarks to the Author):

The authors did a great job in addressing the comments, and it was a pleasure to read the much improved manuscript. I am convinced that this paper makes a valuable contribution to the field and it should therefore be published. I only have a few minor comments:

* The authors write: "This choice [of broad / diverse measures] affords the possibility of identifying the "borders" of self-regulation as a construct or rejecting the discriminant validity of self-regulation in itself." There is already some theorizing related to this in the discussion section, but this point could and should be picked up explicitly. It would be nice if the authors could give a specific answer / conclusion concerning this issue.

We have added to the paragraph beginning with "Finally, this work provides suggestive evidence that psychology should move beyond the idea of a unitary self-regulation construct. ", expanding our discussion to include more than just the predictive work.

* Concerning the two separate psychological spaces the authors argue that "Future reconciliation of these two spaces may be possible." This should also be picked up in the discussion section, and it would be interesting to learn more about the authors' ideas of how one could potentially achieve this.

While we don't have specific prescriptions on how to do this, we added some text arguing that finding such a linking construct should be thought of as a specific objective for some research programs. Find the text at the end of the discussion paragraph beginning with "The breadth of the dataset underlying..."

* Figure 3 (A / C): Instead of showing symmetric correlation matrices (with retest correlations only), it would be helpful if one of the two triangles showed factor-intercorrelations. Moreover, in addition to the colors actual correlation coefficients would be informative.

Thank you for the suggestion. We have included the factor correlations in A/C.

Reviewer #2 (Remarks to the Author):

I served as a reviewer during the previous round of reviews. I was very impressed with the paper and was generally supportive of it being published. My primary concerns rested on a lack of any discussion of the weaknesses of the authors' approach. The present revision now explicitly discusses these issues. Although I might quibble with a few of their minor arguments, I believe the authors have by-and-large adequately addressed my concerns.

Thank you for your former remarks - we're glad we have addressed some of them.

Reviewer #3 (Remarks to the Author):

The authors have largely addressed my suggestions, and I am happy to recommend publication. The contributions of the paper are now much clearer. The Discussion section especially delivers more interesting ideas and fruitful reflection for the field.

I have only two very small remaining suggestions, neither of which are crucial:

1. In the Discussion, in the section in which the limitations of linear bivariate relationships is addressed, it might be useful to concretize this by an example for the reader, with reference to a related psychological literature. For example, I have in mind something like citing the idea that EF may only matter in cases of strong reward sensitivity, etc.

While we would like to replace the abstract example with a more concrete example, as you suggest, we ended up cutting the clarifying "e.g." statement altogether due to space constraints.

2. If there is space, I think the point in response to my previous comment 2E about the eating control surveys being separable from other factors and perhaps being best considered an independent part of a person's psychological phenotype is a good one, and could perhaps be made directly in the paper somewhere (either in the Results or Discussion section).

We have added a statement about the discriminant validity of eating control to the end of the result's paragraph beginning with "To understand the nature of the factors "